# Year-long, broad-band, microwave backscatter observations of an Alpine Meadow over the Tibetan Plateau with a ground-based scatterometer

Jan G. Hofste[1], Rogier van der Velde[1], Jun Wen[2], Xin Wang[3], Zuoliang Wang[3], Donghai Zheng[4], Christiaan van der Tol[1], and Zhongbo Su[1]

[1]Faculty of Geo-Information Science and Earth Observation (ITC), University of Twente, Enschede, Netherlands
[2]College of Atmospheric Sciences, Plateau Atmosphere and Environment Key Laboratory of Sichuan Province, Chengdu University of Information Technology, Chengdu, China
[3]Key laboratory of Land Surface Process and Climate Change in Cold and Arid Regions, Northwest Institute of Eco-Environment and Resources, Chinese Academy of Sciences, Lanzhou, China
[4]National Tibetan Plateau Data Center, Institute of Tibetan Plateau Research, Chinese Academy of Sciences, Beijing, China

**Correspondence:** Jan Hofste (j.g.hofste@utwente.nl)

**Abstract.** A ground-based scatterometer was installed on an alpine meadow over the Tibetan Plateau to study the soil moisture and -temperature dynamics of the top soil layer and air–soil interface during the period August 2017 – August 2018. The deployed system measured the amplitude and phase of the ground surface radar return at hourly and half-hourly intervals over $1-10$ GHz in the four linear polarization combinations (vv, hh, hv, vh). In this paper we describe the developed scatterometer
system, gathered datasets, retrieval method for the backscattering coefficient ($\sigma^0$), and results of $\sigma^0$.

The system was installed on a 5 m high tower and designed using only commercially available components: a Vector Network Analyser (VNA), four coaxial cables, and two dual polarization broadband gain horn antennas at a fixed position and orientation. We provide a detailed description on how to retrieve the backscattering coefficients for all four linear polarization
combinations $\sigma^0_{pq}$, where $p$ is the received- and $q$ the transmitted polarization ($v$ or $h$), for this specific scatterometer design. To account for the particular effects caused by wide antenna radiation patterns ($G$) at lower frequencies, $\sigma^0$ was calculated using the narrow-beam approximation combined with a mapping of the function $G^2/R^4$ over the ground surface. ($R$ is the distance between antennas and the infinitesimal patches of ground surface.) This approach allowed for a proper derivation of footprint positions and -areas, and incidence angle ranges. The frequency averaging technique was used to reduce the effects of fading
on the $\sigma^0_{pq}$ uncertainty. Absolute calibration of the scatterometer was achieved with measured backscatter from a rectangular metal plate and rotated dihedral metal reflectors as reference targets.

In the retrieved time-series of $\sigma^0_{pq}$ for L-band (1.5 – 1.75 GHz), S-band (2.5 – 3.0 GHz), C-band (4.5 – 5.0 GHz), and X-band (9.0 – 10.0 GHz) we observed characteristic changes or features that can be attributed to seasonal or diurnal changes
in the soil. For example a fully frozen top soil, diurnal freeze-thaw changes in the top soil, emerging vegetation in spring, and drying of soil. Our preliminary analysis off the collected $\sigma^0_{pq}$ time-series data set demonstrates that it contains valuable

information on water- and energy exchange directly below the air-soil interface. Information which is difficult to quantify, at that particular position, with in-situ measurements techniques alone.

Availability of backscattering data for multiple frequency bands (raw radar return and retrieved $\sigma^0_{pq}$) allows for studying scattering effects at different depths within the soil and vegetation canopy during the spring and summer periods. Hence further investigation of this scatterometer data set provides an opportunity to gain new insights in hydro-meteorological processes, such as freezing and thawing, and how these can be monitored with multi-frequency scatterometer observations. The data set is available via https://doi.org/10.17026/dans-zjk-rzts (Hofste et al., 2020).

The effects of fading, calibration, and system stability on the uncertainty in $\sigma^0$ are estimated to vary from $\pm\,1.5$ dB for S-band with hh-polarization up to $\pm\,5.5$ dB for C-band with vh-polarization through the campaign. The low antenna directivity (gain) result in additional $\sigma^0$ uncertainty, one that is more difficult to quantify. Estimations point out that it probably will not exceed $\pm\,2$ dB with C-band. Despite these uncertainties, we believe that the strength of our approach lies in the capability of
measuring $\sigma^0$ dynamics over a broad frequency range, $1-10$ GHz, with high temporal resolution over a full-year period.

*Copyright statement.*  TEXT

## 1    Introduction

For accurate climate modelling of the Tibetan Plateau, also known as the 'third pole environment', the transfer processes of
energy and water at the land-atmosphere interface must be understood (Seneviratne et al., 2010; Su et al., 2013). Main quantities of interest are the dynamics of soil moisture and -temperature (Zheng et al., 2017a). Together with sensors embedded into the deeper soil layers, microwave remote sensing is suitable to study these dynamics since it directly probes the top soil layer within the antenna footprint.

A ground-based microwave observatory was installed on an alpine meadow over the Tibetan plateau, near the town of Maqu (China). The observatory consists of a microwave radiometer system called ELBARA-III (ETH L-Band radiometer for soil moisture research) (Schwank et al., 2010; Zheng et al., 2017b), and an microwave scatterometer. Both continuously measure the surface's microwave signatures with a temporal frequency of once every hour year round. The ELBARA-III was installed in January 2016 and is currently still measuring (Su et al., 2020), the scatterometer was installed in August 2017 and continued
to operate until July 2019.

This paper describes the scatterometer system and the dataset that has been collected over the period August 2017 – August 2018 (Hofste et al., 2020). The scatterometer was built with commercially available components: a vector network analyser (VNA), four phase stable coaxial cables, two dual polarization broadband gain horn antennas, and a laptop controlling the scatterometer's operation autonomously. The radar return amplitude and phase were measured over a broad 1- 10 GHz frequency band at all four linear polarization combinations (vv, hv, vh, hh). The scatterometer measured the radar return over a prolonged time with its antennas in a fixed position and orientation, resulting in frequency-dependent incidence angle ranges varying from of $0° \leq \theta \leq 60°$ for L-band (1.625 GHz) to $47° \leq \theta \leq 59°$ for X-band (9.5 Ghz). During the summers of 2017 and 2018 additional experiments were conducted to asses the angular dependence of the backscatter and homogeneity of the local ground surface.

Many other studies exist employing ground-based systems to study microwave backscatter from land. Rather than an airborne- or spaceborne system, ground-based systems allow for high temporal resolution coverage and a high degree of control over the experimental circumstances. Geldsetzer et al. (2007) and Nandan et al. (2016) use specially developed radar systems by ProSensing Inc. to study backscattering from sea ice in the period 2004 - 2011: one system for C- and another for X- & Ku-band. Details on a similar system for S-band can be found in Baldi (2014). The SnowScat system, developed by Gamma Remote Sensing AG (Werner et al., 2010), is another specifically designed scatterometer that operates over 9 - 18 GHz and measures the full polarimetric backscatter autonomously over many elevation- and azimuth angles. SnowScat was used during multiple winter campaigns in the 2009 - 2012 period at two different locations to study the scattering properties of snow layers (Lin et al., 2016). Like in this study, others also designed their scatterometer architecture around a commercially available VNA. For instance, Joseph et al. (2010) used data measured by a truck-based system, operating at C- and L-band, in summer 2002 to study the influence of corn on the retrieval of soil moisture from microwave backscattering. For every band they placed one antenna for transmit and receive on top of a boom. Selection of the individual polarization channels was realized using RF switches. Similar is the University of Florida L-band Automatic Radar System (UF-LARS) (Nagarajan et al., 2014), used by, for example Liu et al. (2016), to measure soil moisture at L-band from a Genie-platform during summer 2012. Another example is the Hongik Polarimetric Scatterometer (HPS) (Hwang et al., 2011), with which microwave backscatter from bean- and corn fields was measured in 2010 and 2013 respectively (Kweon and Oh, 2015). Similar to our study, Kim et al. (2014) used a scatterometer with its antenna in a fixed position and orientation to measure the backscattering during all growth stages of winter wheat at L-, C- and X-band during 2011 - 2012.

The temporal resolution and measurement period covered by the scatterometer data set reported in this paper permits studying both seasonal- and diurnal dynamics of microwave backscattering from an Alpine meadow ecosystem. This in turn allows for investigating the local soil moisture dynamics, the freeze-thaw process, and growth/decay stages of vegetation. Because of the broad frequency range measured (1 – 10 GHz), wavelength-dependent effects of surface roughness and vegetation scattering can be studied as well.

This paper is organized as follows. First, details on the measurement site and the used or existing instruments are given. Followed by details on the scatterometer instrumentation, -setup, -geometries, and -calibration, along with a description of all performed experiments. Next the calculation method for the backscattering coefficient, or normalized radar cross section, $\sigma^0$ is described. Given the system's configuration we show what assumptions and approximation were made for calculating $\sigma^0$ from the measured radar return amplitude- and phase data. We then show some measurement results of $\sigma^0$. These are the angular response of $\sigma^0$ for asphalt, experiments to explore the angular and spatial variability of $\sigma^0$ at the measurement site, and finally some results of the time-series of $\sigma^0$. A list of used symbols can be found at the end of this paper.

## 2  Measurement site

### 2.1  Maqu site

In August 2017 the scatterometer was installed on the tower of the Maqu measurement site (Maqu site) (Zheng et al., 2017b), and operated over the period August 2017 – June 2019. The Maqu site is situated in an Alpine meadow ecosystem (Miller, 2005) on the Tibetan plateau. The site's coordinates are $33°55'$ N, $102°10'$ E, at $3500$ m elevation. The site is located close to the town Maqu of the Gansu province of China.

Besides the scatterometer, other remote sensing sensors placed on the tower are the ELBARA-III radiometer (Schwank et al., 2010) and the optical spectroradiometer system 'Piccolo' (MacArthur et al., 2014), see Fig. 1. The ELBARA-III system has been measuring L-band microwave emission from January 2016 to this date (Su et al., 2020). The Piccolo system measured the reflectance and sun-induced chlorophyll fluorescence of the vegetation over the period July - November 2018.

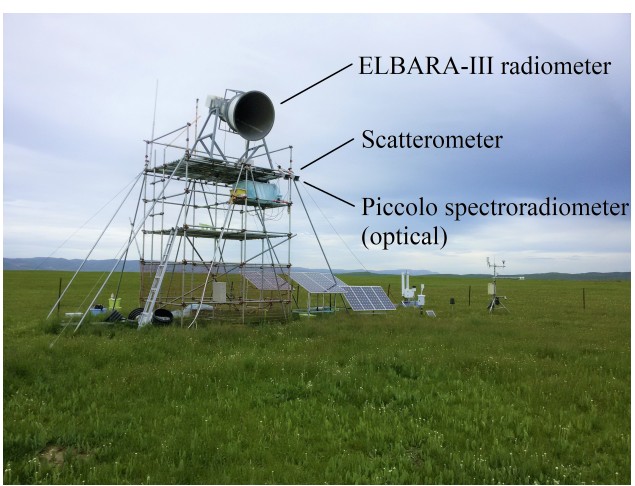

**Figure 1.** Tower of Maqu site containing the scatterometer, the ELBARA-III radiometer, and Piccolo optical spectroradiometer.

## 2.2 Climate

According to Peel et al. (2007) the climate at Maqu is characterized by the Köppen-Geiger classification as 'Dwb', Cold with dry winters. Winter (December - February) and spring (March - May) are cold and dry while the summer (June - August) and autumn (August - November) are mild with monsoon rain.

Figure 2 shows some important hydrometeorological quantities measured at the Maqu site over the period 26 August 2017 – 26 August 2018. Information on used equipment is given in sec 2.4. All shown quantities are also included in the dataset with a temporal resolution of 30 minutes.

From the graphs we observe that the lowest air temperatures $T_{air}$ were measured in January 2018, during which daily minimum values dropped below -20 °C while daily maximum temperatures did not rise above 0 °C. In July – August 2018 $T_{air}$ was highest with maxima above 20 °C.

Soil temperature $T_{soil}$ and soil volumetric liquid water content $m_v$ vary over depth. Depending on the amount of liquid water in the soil the penetration depth of frozen soil at L-band can vary from 10 – 30 cm at the Maqu site (Zheng et al., 2017a). We consider $T_{soil}$ and $m_v$ values at 25 cm depth, which is closest to the maximum aforementioned penetration depth. From the measurements it follows that at 25 cm depth the soil can be considered frozen between 21 December 2017 – 5 April 2018 (arrows in figure). For other depths the freezing- and thawing process is substantially different from the shown curves. During the winter 2018 $T_{soil}$ dropped below 0 °C up to a depth of 70 cm (not shown in Fig.2).

Total precipitation over the considered one-year period was 688 mm. The majority of this amount fell in the months September, October 2017 and in August 2018, while from November 2017 to the middle of March 2018 there was only 7 mm precipitation.

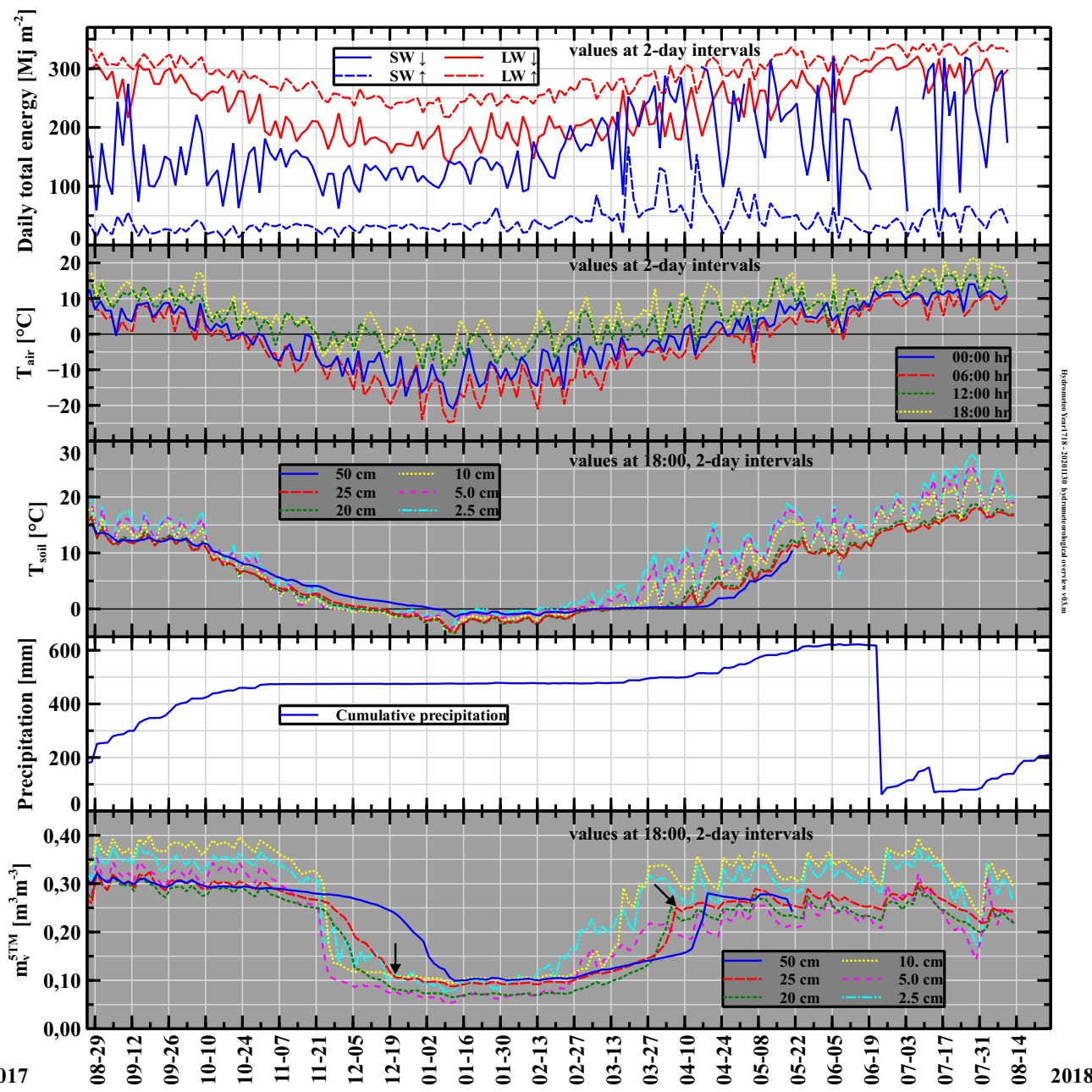

**Figure 2.** Overview of hydrometeorological quantities measured at Maqu site over period 26 August 2017 – 26 August 2018. From top to bottom: Daily total sum of down- and upward hemispherical energy $(\mathrm{Mj\,m^{-2}})$ for short- (285 - 3000 nm) and long (4500 - 40000 nm) wavelengths at two-day intervals, air temperatures (°C) at four times during the day at two-day intervals, soil temperatures $T_{soil}$ (°C) for different depths at two-day intervals, cumulative precipitation mm, and volumetric soil moisture $m_v^{5TM}$ $\mathrm{m^3\,m^{-3}}$ for different depths at two-day intervals. Spatial average volumetric soil moisture $M_v$ is estimated as $M_v = m_v^{5TM} \pm 0.04 \, \mathrm{m^3\,m^{-3}}$.

**Table 1.** Measured vegetation parameters at Maqu-site during summer 2018

|  | 12 July 2018 | 17 August 2018 |
|---|---|---|
| Height (distribution max.) (cm) | 25 | 40 |
| Biomass Fresh ($\mathrm{Kg\,m^{-2}}$) | 0.9 | 1.3 |
| Biomass Dry ($\mathrm{Kg\,m^{-2}}$) | 0.3 | 0.5 |
| VWC (%) | 60 | 62 |
| LAI ($\mathrm{m^2\,m^{-2}}$) | 3.5 | 7 |

## 2.3 Vegetation

The ecosystem classification of the Maqu site is Alpine Meadow according to Miller (2005). The vegetation around the Maqu site consists for a major part of grasses. The growing season starts at the end of April and ends in October, when above-ground biomass turns brown and loses its water. During the growing season the meadows are regularly grazed by lifestock. To prevent the lifestock from entering the site and damaging the equipment a fence is placed around the Maqu site. As a result there is no grazing within the site, causing the vegetation to be more dense and higher than that of the surroundings. Also a layer of dead

plant material from the previous year remains present below the newly emerged vegetation. In Appendix A some photographs are shown of the Maqu site during different seasons, which provide an impression of the site phenology.

To quantify the vegetation cover at the Maqu site, measurements were performed on two days during the 2018 summer: 12 July and 17 August. Vegetation height, above-ground biomass (fresh & over-dried), and leaf area index (LAI) were measured at

135 ten $1.2\times1.2\,\mathrm{m^2}$ sites around the periphery of the 'No-step zone' indicated in Fig. 3. The average quantities over the ten sites are summarized in Table 1. The vegetation height of a single site was determined as the maximum value of the histogram obtained by taking $\geq 30$ readings with a thin ruler at random points within the site area. For each site above-ground biomass and LAI were determined from harvested vegetation within one or two disk areas defined by a 45 cm diameter ring. Immediately after harvest all biomass was placed in air-tight bags so that the fresh- and dry biomass could be determined by weighing the bag's

content before and after heating with an oven. The LAI was determined immediately after harvest with part of the harvested fresh biomass by the plotting method described in He et al. (2007).

## 2.4 Hydrometeorological sensors

Table 2 lists all hydrometeorological instruments used for this study along with their reported measurement uncertainties. Air temperature was measured with a Platinum resistance thermometer, type HPM 45C, installed 1.5 m above the ground and pre-

145 cipitation (both rain and snow) was measured with a weight-based rain gauge, type T-200B. The depth profile of volumetric soil moisture $m_v$ ($\mathrm{m^3\,m^{-3}}$) was measured with an array of 20 capacitance sensors, type 5TM, that were installed at depths ranging from 2.5 cm to 1 m (Lv et al., 2018). All sensors in the array are also equipped with a thermistor, enabling the measurement of the soil temperature depth profile $T_{soil}$ (°C). The soil moisture and -temperature was logged every 15 minutes for the period

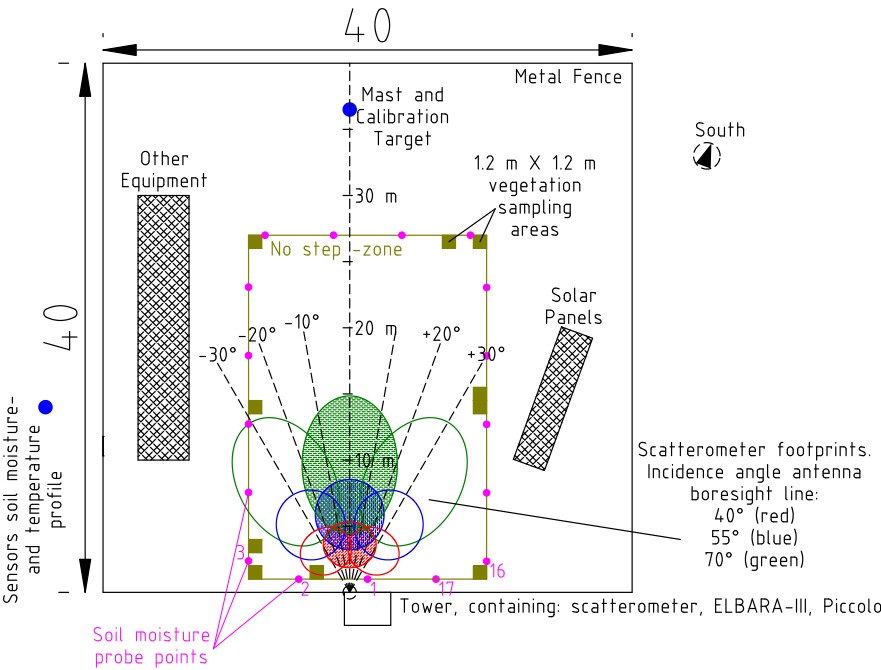

**Figure 3.** Map of the Maqu site. Scatterometer footprints for C-band with vv polarization shown for different incidence angles of antenna boresight line: $\alpha_0 = 40, 55, 70°$. Also shown are antenna azimuth angles $\phi$.

**Table 2.** Overview of relevant hydrometeorological sensors Maqu site

| Quantity | Type, Manufacturer: | Unit, Uncertainty: |
|---|---|---|
| Volumetric soil moisture $m_v$ | 5TM, Meter Group | $\pm 0.02\,\mathrm{m^3\,m^{-3}}$ (Zheng et al., 2017b) |
| Volumetric soil moisture $m_v$ | ThetaProbe ML2x, Delta-T Devices | $\pm 0.05\,\mathrm{m^3\,m^{-3}}$ |
| Soil temperature | 5TM, Meter Group | $\pm 1\,°\mathrm{C}$ |
| Air temperature | HPM 45C, Campbell Scientific | $\pm 1\,°\mathrm{C}$ |
| Precipitation (rain & snow) | T-200B, Geonor | $\pm 0.6\,\mathrm{mm}$ |
| Short- and long wave up- and downward irradiance | NR01, Hukseflux | $\pm 5\%\,\mathrm{W\,m^{-2}}$ |

of August 2017 – August 2018 with Em50 data loggers (manufacturer: Meter Group) that were buried nearby with the sensors.

The location of the buried sensor array is indicated in Fig. 3.

We estimate that the spatial average top soil moisture content over the Maqu site $M_v$ $(\mathrm{m^3\,m^{-3}})$ is linked to $m_v$ as measured by the 5TM sensors at 2.5 and 5 cm depth $(m_v^{5TM})$ according to

$$M_v = m_v^{5TM} \pm S_{tot} \tag{1}$$

where $S_{tot}$, with value 0.04 $\mathrm{m^3\,m^{-3}}$, is the total standard deviation of spatially measured $m_v$ with a hand held impedance probe, type ThetaProbe ML2x. Refer to Appendix B for additional information.

## 3 Scatterometer and its operation

### 3.1 Instrumentation

The main components of the scatterometer are a 2-port vector network analyser (VNA), type PNA-L 5232A (manufacturer:
Keysight), four 3 m long phase stable coax cables, type Succoflex SF104PEA (manufacturer Huber + Suhner), and two dual polarization broad band horn antennas, type BBHX9120LF (manufacturer: Schwarzbeck). The test-port couplers of the VNA are omitted and the coax cables are connected according to the schematic in Fig. 4 (Agi). This configuration allows for mea-

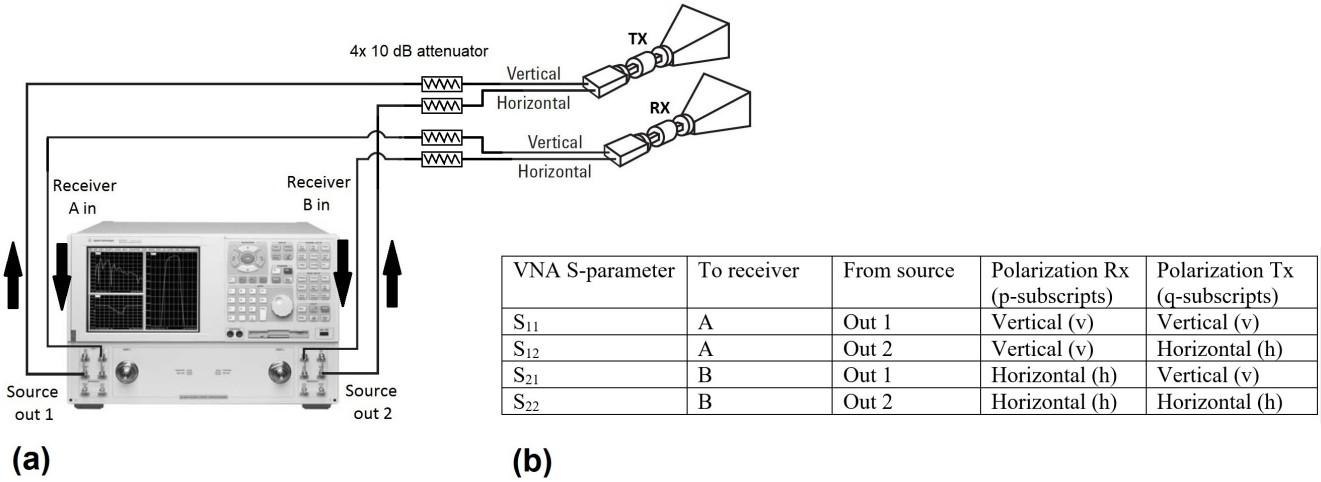

| VNA S-parameter | To receiver | From source | Polarization Rx (p-subscripts) | Polarization Tx (q-subscripts) |
|---|---|---|---|---|
| $S_{11}$ | A | Out 1 | Vertical (v) | Vertical (v) |
| $S_{12}$ | A | Out 2 | Vertical (v) | Horizontal (h) |
| $S_{21}$ | B | Out 1 | Horizontal (h) | Vertical (v) |
| $S_{22}$ | B | Out 2 | Horizontal (h) | Horizontal (h) |

(a)   (b)

**Figure 4.** Connection scheme of scatterometer and correspondence S-parameters to polarization channels for transmit (Tx) and receive (Rx). (a) Both dual polarization broadband antennas, one for Tx, the other for Rx, are connected to the VNA as indicated (Agi). Arrows indicate direction of signal. (b) Overview correspondence of four VNA S-parameters to the four polarization channels.

suring all four polarization channels: vv (transmit in vertical direction, receive in vertical direction), vh, vh, and hh. Between all four coaxial cables and their respective VNA connectors 10 dB attenuators, type SMA attenuator R411.810.121 (manufac-
165 turer: Radiall) were inserted to prevent interference from internal reflections travelling multiple times up- and down the coaxial cables. Measurements are performed by instructing the VNA to measure the four scattering parameters (S-parameters)[1] ($-$) over a stepped frequency sweep 0.75 – 10.25 GHz. Given the aforementioned connection scheme the correspondence between recorded S-parameters and transmit- /receive polarization channels are as indicated in Fig. 4b. Note that the VNA software, by default, accounts for the test-port couplers by adding 16 dB to the signal measured by receivers A and B. To protect the VNA
from weather it is placed inside a water proof enclosure equipped with fans to provide air ventilation. The antenna radiation

---

[1]Not to be confused with the scattering amplitudes used in scattering theory, which have units m, see for example Ulaby and Long (2017).

patterns are measured in the principal planes by the manufacturer over the 1 – 10 GHz band (Schwarzbeck Mess-Elektronic, 2017). As a summary, the full width half maximum (FWHM) intensity beamwidths over frequency are shown in Appendix D, Fig. D1. The scatterometer is placed on a tower as shown in Fig. 1. The two antenna apertures are at a distance approximate $H_{ant} = 5$ m above the ground ($H_{ant}$ depends on the antenna boresight angle $\alpha_0$) and are separated from each other horizontally by $W_{ant} = 0.4$ m.

Deployed reference targets to calibrate the scatterometer were a rectangular plate and two dihedral reflectors. The rectangular plate reflector was constructed from light-weight foam board covered with 100 μm aluminium foil and had frontal dimensions $a = 85\,\text{cm} \times b = 65$ cm. A small dihedral reflector was constructed from steel, its frontal dimensions were $a = 57\,\text{cm} \times b = 38$ cm. A second large dihedral reflector was also constructed with foam board and aluminium foil, its frontal dimensions were $a = 120\,\text{cm} \times b = 65$ cm. A height-adjustable metal mast was used to position the reference targets. To minimize reflection from this mast it was covered by pyramidal absorbers, type 3640-300 (manufacturer: Holland Shielding), having a 35 dB reflection loss at 1 GHz under normal incidence.

### 3.2 Setup

Figure 5 shows all relevant geometries for the performed experiments. The two antenna apertures are at distance $H_{ant}$ above the ground surface. The separation between the two antenna apertures $W_{ant} = 0.4$ m is small compared to the target distance (ground or calibration standards) which justifies using the geometric centre of the two apertures for all calculations. Every area segment $dA$ (m$^2$) of the ground surface has its own distance to the antennas $R$ and angle of incidence $\theta$. Angles $\alpha$ and $\beta$ are angular coordinates of R. Angle $\alpha$ is defined between the tower's vertical axis and the orthogonal projection of the line from antennas to a ground surface segment onto the plane formed by the tower's vertical axis and the antenna boresight direction line. Angle $\beta$ is defined between line from antennas to a ground surface segment and projection of that same line onto the plane formed by the tower's vertical axis and the antenna boresight direction line. The planes in which $\alpha$ and $\beta$ lie are also the antenna's principal planes (see for example (Balanis, 2005)). For the antenna boresight direction $\alpha = \alpha_0$ and $\beta = \beta_0$. The antenna rotation around the tower's vertical axis is defined as azimuth rotation $\phi$.

According to Bansal (1999) the antenna's far field distances $R_{ff}$ (m) are linked to the antenna's largest aperture dimension $D$ (m) and wavelength $\lambda$ via

$$R_{ff} \geq \begin{cases} 5D & : \frac{1}{3} \leq \frac{D}{\lambda} \leq \frac{5}{2} \\ \frac{2D^2}{\lambda} & : \frac{5}{2} < \frac{D}{\lambda} \end{cases} \tag{2}$$

The antenna aperture is rectangular with dimension $D = 0.2$ m, which leads to $R_{ff} \geq 1$ m for 1 - 3.5 GHz and $R_{ff} \geq 2.7$ m for 3.5 - 10 GHz. Given that with all measurements the distance to the ground surface is larger than 2.7 m the radiation patterns as measured by the manufacturer apply (Schwarzbeck Mess-Elektronic, 2017).

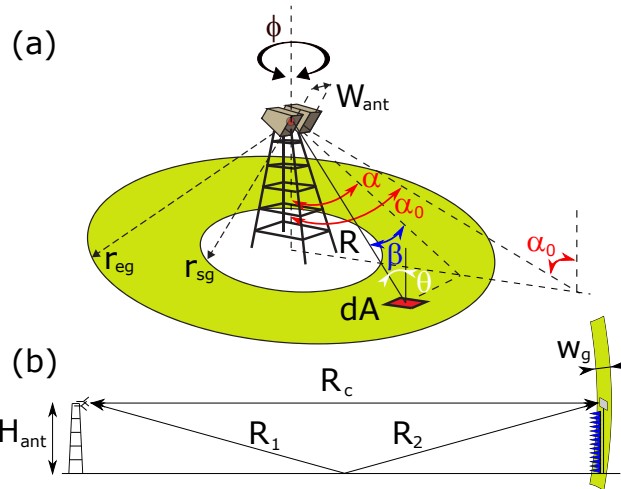

**Figure 5.** Schematic of scatterometer geometry. (a) Every infinitesimal area $dA$ has its own distance $R$ to the geometric centre between antenna apertures (red dot) and angle of incidence $\theta$. Angles $\alpha$ and $\beta$ lie within the antennas principal planes, $\alpha_0$ denotes the angle of antenna boresight. The green ring is a projection of the spherical gating shell with radii $r_{sg}$ and $r_{eg}$ onto the ground. (b) Side view of geometry during measurement of reference standards. Green ring depicts cross section of spherical gating shell with width $w_g$.

The radar return from the rectangular metal plate reference target was used to calibrate the scatterometer for the co-polarization channels, as illustrated in Fig. 5(b). The two metal dihedral reflectors were used as depolarizing reference targets (Nesti and Hohmann, 1990) to calibrate the cross-polarization channels. We used two dihedrals, measured at different distances $R_0$, in order to satisfy additional requirements. Refer to Appendix C for the measurement details and validation-exercise results.

Time-domain filtering, or gating, was used as part of post processing to remove the antenna-to-antenna coupling and undesired scattering contributions from the radar return signal for both the reference target- and the ground-return measurements. The ring on the ground surface in Fig. 5 is the intersection of a spherical shell with radii $r_{sg}$ and $r_{eg}$ centred at the antennas and the ground surface. It represents the selected ground surface area for the gating algorithm: roughly put, scattering returns from features within the spherical shell remain in the radar return signal while those outside the shell are removed. The application of gating with VNA-based scatterometers is described in more detail in for example (Jersak et al., 1992) or (De Porrata-Dória i Yagüe et al., 1998). Details on our gating process and related peculiarities regarding our scatterometer can be found in Appendix D.

### 3.3 Experiments

During all experiments, VNA measurements were performed with a stepped 0.75 – 10.25 GHz frequency sweep at 3 MHz resolution (3201 points). The dwell time per measured frequency was 1 μs, equivalent to a two-way travelling distance for the microwave signal of 150 m. The intermediate-frequency (IF) bandwidth was minimized to 1 KHz to increase the signal-to-

noise ratio.

In this paper, we describe the following experiments: a measurement of the $\sigma^0$ for asphalt at various $\alpha_0$ angles, measurements of $\sigma^0$ for different $\alpha_0$- and $\phi$ angles at the Maqu site, and finally the measurement of $\sigma^0$ over a one-year period. Table 3 summarizes the experiment geometries and dates of execution. With the angular-variation experiments the scatterometer antennas were

mounted on a motorized rotational stage. Depending on the angle $\alpha_0$, $H_{ant}$ would vary according to $H_{ant} = H_0 - 0.5\cos(\alpha_0)$, with $H_0 = 2.95$ or $5.2$ m for the asphalt- or Maqu experiments respectively. With the time-series experiment the antennas were fixed on a tower rod, such that $\alpha_0$ was 55 °. All angular-variation experiments were conducted within one afternoon. With the time-series experiment the radar return was measured either once or twice per hour continuously.

**Table 3.** Overview scatterometer experiments

|  | Date: | $\phi$ (°): | $\alpha_0$ (°): | $H_{ant}$ (m): |
|---|---|---|---|---|
| Angular variation $\sigma_0$ asphalt | 4 May 2017 | 00 | 35 40 .. 75 | 2.55 2.55 .. 2.80 |
| Angular variation $\sigma_0$ Maqu | 25 August 2017 | -20 -15 -10 -05 00 +10 +15 +20 | 35 40 .. 70 | 4.80 4.80 .. 5.05 |
| Angular variation $\sigma_0$ Maqu | 29 June 2018 | -30 -20 -15 -10 -05 00 +05 +10 +20 +25 +30 | 35 40 .. 70 | 4.80 4.80 .. 5.05 |
| Angular variation $\sigma_0$ Maqu | 19 August 2018 | -30 -20 -10 00 +10 +20 +30 | 35, 55, 70 | 4.80 4.90 5.05 |
| Time series $\sigma_0$ Maqu | 26 August 2017 – 26 August 2018 | 00 | 55 | 4.70 |

# 4    Derivation of the backscattering coefficient

## 4.1    Effects of wide radiation patterns

The power received by a monostatic radar- or scatterometer system from a distributed target with backscattering coefficient $\sigma_{pq}^0(\theta)$ (m² m⁻²) is given by the radar equation (Ulaby et al., 1982)

$$P_p^{RX} = \frac{\lambda^2}{64\pi^3} P_q^{TX} G_0^2 \int \frac{G^2}{R^4} \sigma_{pq}^0(\theta).dA \tag{3}$$

where it is assumed that the same antenna is used for both transmitting (Tx) and receiving (Rx). $P_q^{Tx}$ is the transmitted-, and

$P_p^{Rx}$ the received power respectively (W). The subscripts of the powers refers to the linear polarization directions: horizontal h, or vertical v. With $\sigma_{pq}^0$ the first subscript refers to the polarization direction of the scattered- and the second to that of the incident wave. $G$ (−) denotes the normalized angular gain pattern of the antenna with peak value $G_0$ (−). Equation 3 represents an ideal lossless system, in practice any scatterometer has frequency dependent losses or other signal distortions.

These frequency dependent phase- and amplitude modulations can be accounted for by measuring the radar return of a reference
target $P_p^c$ with known radar cross section (RCS) $\sigma_{pq}$ (m$^2$) (see Appendix. C) and subsequently using this to calibrate the system.
This procedure is often referred to as external calibration. Substitution of terms associated with the reference measurement into
Eq. 3 leads to

$$P_q^{RX} = P_q^c \frac{(R_0)^4}{\sigma_{pq}} \int \frac{G^2}{R^4} \sigma_{pq}^0(\theta).dA \tag{4}$$

where $R_0$ (m) is the distance at which the reference target was measured. In the case of a scatterometer with narrow beamwidth
antenna, all integrand terms of Eq. 4 can be approximated as being constants, the so-called 'narrow-beam approximation'
(Wang and Gogineni, 1991), so that we obtain

$$P_p^{RX} = P_p^c \frac{(R_0)^4}{\sigma_{pq}} \frac{1}{(R_{fp})^4} \sigma_{pq}^0(\theta) A_{fp} \tag{5}$$

where $A_{fp}$ is the scatterometers 'footprint', notably the area (m$^2$) for which the surface projected antenna beam intensity is
equal to or larger than half its maximum value. $R_{fp}$ (m) refers to the distance between the antenna and footprint centre.

For this dataset $\sigma_{pq}^0(\theta)$ is estimated by employing Eq. 5 in combination with a mapping of the term $G^2/R^4(x,y)$ from Eq.
4 over the ground surface. Due to the wide antenna radiation patterns, especially with low frequencies, the area that is to be
associated with the measured scatterometer signal, i.e. the footprint is typically not located where the antenna boresight line
intersects the ground surface. Instead the footprint appears closer to the tower base. Figure 6 demonstrates this effect for the
case of 5 GHz at $\alpha_0 = 55$ °. Shown is the mapping over the ground surface of the $G^2/R^4$ -term from Eq. 4. This footprint-shift
effect is strongest with the widest antenna radiation patterns (thus with low frequencies) and for large $\alpha_0$ angles. The footprint

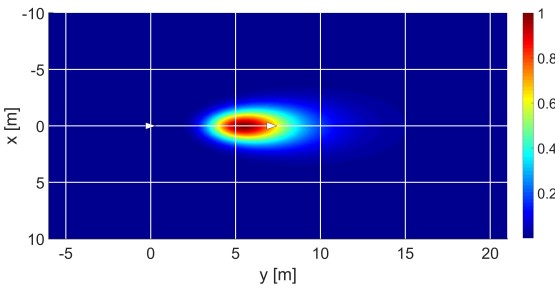

**Figure 6.** Example of $G^2/R^4(x,y)$ with Gaussian antenna radiation patterns. Plot normalized to its peak value. $x$ and $y$ are ground surface
coordinates. White triangle at coordinate (0,0) represents the tower location and other white triangle indicates intersection point of the antenna
boresight line and the ground surface. $\alpha_0 = 55°$, $f = 5$ GHz and polarization is vv.

position and dimensions were found using the mapping $G^2/R^4(x,y)$ over the ground surface. The applied criterion was that
the footprint contains 50% of the total projected intensity onto the ground surface. After the footprint edges were defined the
incidence angle ranges were derived from them using straightforward trigonometry.

Because of the low directivity (gain) of the antennas and the unknown nature of $\sigma_{pq}^0$ over $\theta$, there is an inherent uncertainty in the absolute level of our retrieved $\sigma_{pq}^0$ values (for a certain $\theta$ range). Quantifying this uncertainty is outside the scope of this paper. In Sec. 5.1.2 we do however provide an estimate of what this uncertainty could be. Despite this limitation we show that nevertheless the temporal dynamics of $\sigma_{pq}^0$, for various wavelengths, are captured by the $\sigma^0$ dataset retrieved with our system. Alternatively, when using this dataset together with a microwave scattering model the low directionality issue can be resolved by using the measured radar return $P_p^{Rx}$ instead of the derived $\sigma^0$ presented here. The angle-dependent $\sigma_{pq}^0(\theta)$ then may be obtained by the microwave scattering model and simply applied in Eq. 4 to simulate the radar return, which then can be compared to the measured $P_p^{Rx}$ in this dataset.

## 4.2 Implementation of the radar equation

We rewrite Eq. 5 so that the backscattering coefficient of the surface $\sigma^0$ ($\mathrm{m^2\,m^{-2}}$) is related to the average received backscattered intensity $\bar{I}$ ($\mathrm{Wm^{-1}}$) as (Ulaby and Long, 2017)

$$\sigma^0 = K^{-1}\bar{I} \tag{6}$$

where for brevity the polarization subscripts are omitted. The factor $K$ ($\mathrm{W\,m^{-1}}$) is a constant for the bandwidth considered given by

$$K = \frac{\lambda^2}{4\pi^3} I^t \frac{G^2}{R_{fp}^4} A_{fp} \tag{7}$$

where $I^t$ ($\mathrm{W\,m^{-2}}$) is the transmitted intensity by the scatterometer. For all terms in $K$ the centre frequency is used. Similar as with Eq. 4, we can substitute $I^t$ in Eq. 7 by the relevant radar parameters when a reference target is measured, yielding

$$K = \frac{1}{2} c\epsilon_0 (E_0^{g0} - E_{b0}^{g0} - E_b)^2 \frac{G(\alpha,\beta)^2}{G(\alpha_0,\beta_0)^2} \left(\frac{R_0}{R_{fp}}\right)^4 \frac{A_{fp}}{\sigma} = \frac{1}{2} c\epsilon_0 (E_0^{g0} - E_{b0}^{g0} - E_b)^2 \left(\frac{R_0}{R_{fp}}\right)^4 \frac{A_{fp}}{\sigma} \tag{8}$$

$E_0^{g0}$ ($\mathrm{V\,m^{-1}}$) is the measured backscattered field from the reference target (subscript 0 represents 'reference') and $E_{b0}^{g0}$ ($\mathrm{V\,m^{-1}}$) is the measured background level during calibration, i.e. the measured backscattered electric field when the calibration standard was removed from the mast while the pyramid absorbers remained in place. With both terms the superscript $g0$ (for 'gate' during reference measurements) indicates that an identical gate was used. The field strength associated with the minimum signal level measurable with the scatterometer is denoted $E_b$. The prefactors light speed $c$ ($\mathrm{m\,s^{-1}}$) and the permittivity of vacuum $\epsilon_0$ ($\mathrm{F\,m^{-1}\,m^{-1}}$) convert the electric field strengths into time-average intensity. In the middle part of Eq. 8 the antenna gain functions are written explicitly. $G(\alpha,\beta)$ represents the antenna gain functions when measuring the ground return, while $G(\alpha_0,\beta_0)$ represents the situation when the radar return of the reference targets is measured. When using the narrow beam approximation (Eq. 5) and when the reference target is aligned to the antenna boresight direction the fraction becomes unity and the right part of Eq. 8 follows. The middle part is used in Appendix. E2.1 when alignment uncertainty of the reference targets is discussed.

In the context of Rayleigh fading statistics with square-law detection (Ulaby et al., 1988), the average received intensity $\bar{I}$ ($\mathrm{W\,m^{-2}}$) is linked to $I_N$ ($\mathrm{W\,m^{-2}}$), which is the measured intensity averaged over $N$ independent samples ($N$ footprints or $N$

frequencies), according to

$$\bar{I} = \frac{I_N}{1 \pm 1/\sqrt{N}} \tag{9}$$

Note that $\bar{I}$, like $\sigma^0$ is an implied ground surface property. The quantity that is actually measured, $I_N$, is an estimator for $\bar{I}$. Equation 9 holds for $N \geq 10$, since then the probability density function of $I_N$ approaches a Gaussian distribution (Ulaby et al., 1982) according to the central limit theorem. The denominator in Eq. 9 represents a 68% confidence interval ($\pm 1$ standard deviation) for $\bar{I}$. More details on fading are described in Section 4.3.

In turn, $I_N$ is calculated from the measured backscattered electric field from the ground target incident on the receiving antenna $E_e^g$ (V m$^{-1}$) by

$$I_N = \frac{1}{2} c \epsilon_0 \frac{1}{N} \sum_{n=1}^{N} (E_e^g(f_n) - \langle E_{cr}^g \rangle - E_b)^2 \tag{10}$$

The subscript $e$ denotes 'envelope' magnitude of the complex signal, as in (Ulaby et al., 1988)[1] and the superscript $g$ indicates that the signal is gated. $E_{cr}^g$ (V m$^{-1}$) is an offset formed by part of the signal transmitted from the transmit antenna coupling directly into the receive antenna (antenna cross coupling). Although the majority of this coupling can be filtered out by using time-domain gate filtering a remnant is still present (hence 'coupling remnant' in the subscript) and must be accounted for. The bandwidth-average magnitude of $E_{cr}^g$ is to be subtracted from the received signal (Appendix E3). Note that the exact gate is applied as with $E_e^g$. The last term $E_b$ represents the minimum detectable signal. A similar form of offset subtraction from the measured radar return $E_e^g$ was done in Nagarajan et al. (2014).

## 4.3  Fading and bandwidth selection

Fading is the phenomena that radar return of a distributed target with uniform electromagnetic properties has varying magnitudes and phases when different locations or slightly different frequencies are measured (Ulaby et al., 1988), (Monakov et al., 1994). To remove this varying nature from a surface-classifying quantity like $\sigma_{pq}^0$ averaging must be performed. By definition $\sigma_{pq}^0$ is the average radar cross section of a certain type of distributed target, e.g. forest, asphalt, wheat field, normalized by the illuminated physical surface area. $\sigma^0$ is proportional to the average measured received power $P^{Rx}$ (Eq. 5) or intensity $\bar{I}$. Therefore, determining $\bar{I}$ and $\sigma^0$ requires $N$ statistically independent samples so that the sample average $I_N$ approaches the actual average $\bar{I}$ proportionally to $1/\sqrt{N}$ in accordance with the central limit theorem.

Practically, this can be done either by measuring $I$ at $N$ different locations over the surface, called spatial averaging, or with the frequency averaging -technique (see for example (Ulaby et al., 1988)). With the latter, physical properties governing the scattering, permittivity and surface roughness are considered frequency invariant over a certain bandwidth. Subsequently, $N$

---

[1]In reality the measured fields or signals remain complex until after the gating process. We however stick to this terminology for clarity.

different frequencies should be selected according to some criteria that accounting for fading. Both averaging techniques can be used simultaneously as done by Nagarajan et al. (2014) to increase the total number of independent samples. We solely applied the frequency-averaging technique because during the time-series measurements our antennas were in a fixed position and orientation. We assumed the single footprint area to be representative for the whole surface of the Maqu site. In Sec. 5.2.2 we show this assumption is justified. The used method for finding the number $N$ of statistically independent samples within a bandwidth $BW$ is described in Mätzler (1987):

$$N = \frac{2BW\Delta R}{c} \tag{11}$$

where $\Delta R = r_{sg} - r_{eg}$. Subsequently, with $N-1$ intervals of $\Delta f$ (Hz), $N$ frequencies are selected from within $BW$.

As indicated above, with the application of the frequency averaging technique it is assumed that the backscatter behaviour across the selected $BW$ is uniform. To assess the validity of this assumption for bare surface, the improved integral equation method ($I^2$EM) surface scattering model (Fung et al., 2002) is applied using the roughness parametrization reported in Dente et al. (2014) and a (frequency dependent) effective dielectric constant $\epsilon_{soil}(f)$ according to the dielectric mixing model by Dobson et al. (1985).

Over a $BW$ the mean value $\langle \sigma^0(BW) \rangle$ is calculated, followed by the ratios $\sigma^0(BW_{lo})/\langle \sigma^0(BW) \rangle$ and $\sigma^0(BW_{hi})/\langle \sigma^0(BW) \rangle$ to quantify the change of $\sigma^0$ over the $BW$. In general the $I^2$EM model predicts that the change is largest for long- and smallest for short wavelengths and that it is largest for hh polarization and smallest for vv polarization. Furthermore, the RMS surface height is the most sensitive target parameter. As an example, figure 7 shows the calculation result for hh polarization with a

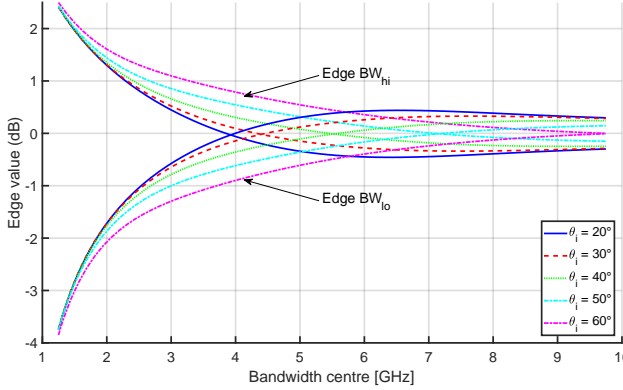

**Figure 7.** Variation of $\sigma^0_{hh}$ per $BW$ calculated with combined $I^2$EM- (Fung et al., 2002) and Dobson (Dobson et al., 1985) model. Horizontal axis shows centre frequency of bandwidth $BW = 0.5$ GHz. Curves indicate the values (in dB) to be added to $\langle \sigma^0_{hh}(BW) \rangle$ at edges of $BW$ for different $\theta$ angles. Shown calculation uses: $s = 1$ cm, $\ell = 10$ cm, $m_v = 0.25 \, \mathrm{m^3 \, m^{-3}}$, and $T_{soil} = 15\,^{\circ}$C.

$BW$ of 0.5 GHz. From the graph we can read that for a centre frequency of 2.75 GHz that the retrieved $\sigma^0_{hh}$ for that $BW$ can be expected to vary $+1.0$ to $-1.2$ dB for $\theta = 50°$.

Based on the above calculations we chose $BW = 0.25$ GHz for L-band, $BW = 0.5$ GHz for S- & C-band, and $BW = 1.0$ GHz for X-band. These bandwidths will lead to $N$-values around 10 which is sufficient to let the probability density function

of $I_N$ approach a Gaussian distribution, as explained in Sec. 4.4. Further increment of $BW$ was considered not to outweigh the loss of frequency resolution, especially at S-band.

## 4.4   Procedure

In Figure 8 the procedure for deriving the backscattering coefficient is depicted. The different steps indicated in the figure are explained here:

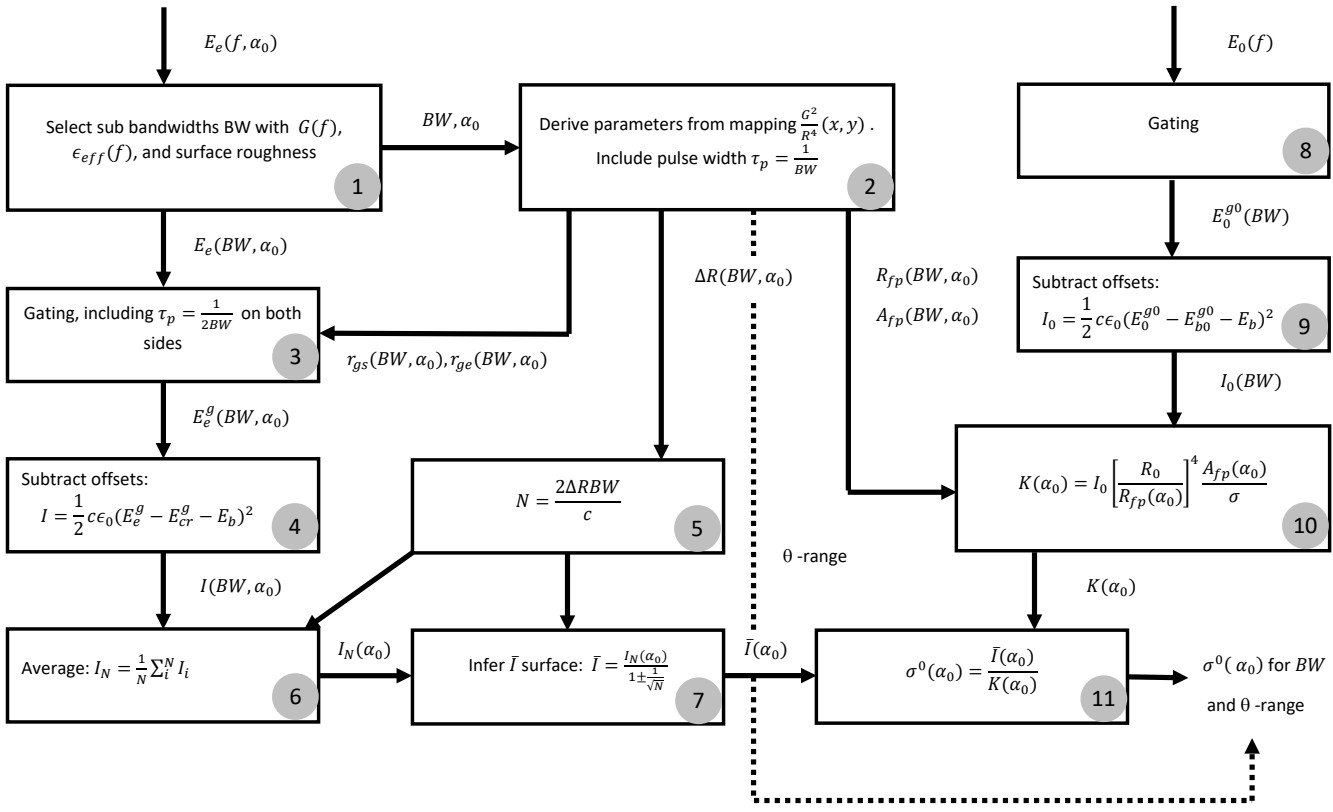

**Figure 8.** Flowchart of $\sigma^0$ derivation process. Inputs are the measured backscattered electric fields of the surface target $E_e(f, \alpha_0)$ and the calibration standard $E_0(f)$. The process follows from 1 to 11 in sequence.

1. We start with $E_e$ measured over the full $0.75 - 10.25$ GHz band at angle $\alpha_0$: $E_e(f, \alpha_0)$. Bandwidths $BW$ are selected based on the change of $G(\alpha, \beta)$ over frequency (Appendix D), the number of independent frequency samples $N$ that may be retrieved from $BW$, and the estimated change of backscattering properties over frequency of the ground surface as is discussed in Sec. 4.3. Result is the bandwidth selection $E_e(BW, \alpha_0)$.

2. With $BW$ and $\alpha_0$ as input, $G^2/R^4(x,y)$ is mapped for all frequencies within $BW$ using the antenna radiation patterns measured by the manufacturer. The region associated with 50 % of the total projected intensity onto the ground is determined to set appropriate gating times, or distances $r_{gs}$, $r_{ge}$, and for calculating the $A_{fp}$, $R_{fp}$, and the $\theta$ range. Half the pulse width $c/(2BW)$ is subtracted from $r_{sg}$ and added to $r_{eg}$, quantities $A_{fp}$, $R_{fp}$, and the $\theta$ range are changed accordingly.

3. The gate is applied to $E_e(BW,\alpha_0)$, resulting in the gated backscattered field $E_e^g(BW,\alpha_0)$.

4. The coupling remnant $E_{cr}^g(BW)$ and minimal detectable signal $E_b$ (a constant) is subtracted from $E_e^g(BW,\alpha_0)$ for each measured frequency. The result is squared and converted into intensity $I(BW,\alpha_0)$.

5. The number of statistically independent frequency samples $N$ within $BW$ is calculated with $\Delta R = r_{eg} - r_{sg}$ (Sec. 4.3).

6. From the $I(BW,\alpha_0)$ spectrum $N$ intensities are selected at equidistant intervals of $\Delta f = BW/N - 1$ and averaged to $I_N(\alpha_0)$.

7. With $I_N(\alpha_0)$ and $N$, $\bar{I}(\alpha_0)$ is calculated using Eq. 9. The denominator $1 \pm 1/\sqrt{N}$ implies that $\bar{I}$ is estimated with a 68 % confidence interval.

8. The gated backscattered signal from the reference target $E_0^{g0}(BW)$ is determined for the full $0.75 - 10.25$ GHz band under the assumption that $G \approx 1$ for all frequencies (see Appendix D). After gating the relevant $BW$ of $E_0^{g0}$ is selected.

9. The measured response from the mast without reference target $E_{b0}^{g0}(BW)$ is subtracted from the reference target response. Subscript $b0$ denotes background calibration, the superscript $g0$ indicates that the same gate was used as with the reference target response. Also $E_b$ is subtracted. The result is squared and converted into intensity $I_c(BW)$.

10. The $I_c(BW)$ is used to calculate the factor $K$, given the footprint area $A_{fp}$ and centre distance $R_{fp}$ (Eq. 7).

11. The final step is the application of Eq. 6 with $\bar{I}(\alpha_0)$ and $K(\alpha_0)$ as inputs to obtain $\sigma^0$. By steps 2 and 6 the derived $\sigma^0$ is to be associated with the chosen $BW$ and calculated $\theta$ -range. By step 7 a 68 % confidence interval applies to $\sigma^0$.

# 5 Measurement results

## 5.1 Measurement uncertainty

### 5.1.1 Fading- and systematic measurement uncertainty

Besides uncertainty due to fading, systematic measurement uncertainty was also considered in the retrieval of $\sigma^0$. The radar returns and subsequent $\sigma^0$ -values derived from it have a systematic measurement uncertainty whose main contributors are the temperature-induced radar return uncertainty $\Delta E_T$ (V m$^{-1}$) and reference target measurement uncertainty $\Delta K$ (in dB, as

is relative value). For both factors we estimate their respective uncertainty levels (see Appendix E1 and Appendix E2 respectively) and how these propagate into an overall $\sigma^0$ measurement uncertainty together with the fading uncertainty. In this context we also consider the system's offsets levels formed by the antenna-to-antenna coupling remnant $\langle E_{cr}^g(f) \rangle$ (in $\mathrm{V\,m^{-1}}$, averaged over $BW$) and the minimal signal strength measurable by the VNA, or background $E_b$ ($\mathrm{V\,m^{-1}}$). The former is derived from measurements with the antennas aimed skywards. From $E_b$ the minimal measurable RCS (given a certain distance $R$ to target) $\sigma_{min}$ can be calculated via Eq. 5, where instead of the product $\sigma^0 A_{fp}$ a RCS value is to be calculated using the power levels associated with $E_b$.

Table 4 lists all aforementioned quantities per $BW$ and polarization channel. The uncertainty $\Delta K$ and $\sigma_{min}$ values are shown as is, but for the other quantities the resulting receiver power levels (in dBm) are shown to allow for comparison with other systems. As explained in sec. 3.1 the VNA actually measures the four S-parameters which are the (complex) ratios of the received- over the transmitted wave voltage for the four polarization channels. The received wave voltages are proportional to the different electric field strengths $E_e$, $E_0$, etc. described in sec. 4.2. The transmitted wave voltage, or actually its power, is constant at 10 dBm with all measurements. For the calculation of $\sigma^0$ by Eq. 6 it is irrelevant whether the electric field strengths, wave amplitudes or S-parameter magnitudes are used since the transmission-related components and/or prefactors simply cancel out. Conversion from measured S-parameters (which are associated with the corresponding scattered electric field strengths) to receiver power is done by subtracting -16 dB, which was added by the VNA software to account for the test-port coupler, and adding 10 dBm. As an example we consider a ground measurement taken on 2017-12-24 00:10:00. The VNA measured $dB(S_{11}) = -85.24$ dB for 2.8 GHz (S-band) with vv polarization. The power at the VNA receiver then was $-85.24 - 16 + 10 = -91.24$ dBm.

As Table 4 shows, the received power associated with $\Delta E_T$ and $\langle E_{cr}^g \rangle$ are, in general, highest for L- and lowest for X-band. Also, the cross polarization channels have lower values than those for co polarization. As for $\Delta E_T$, we do not have a clear explanation for this behaviour. For $\langle E_{cr}^g \rangle$ we argue that the L-band values are highest due to the stronger coupling because of the broadest radiation patterns at that band. The co- values are higher than with cross polarization because of how the electric-field lines allow for better coupling with the former (see Appendix E3). The power levels associated with $E_b$ were derived from the specifications documentation of the VNA (Keysight, 2018). The 'typical' receiver noise levels described therein are specified for a 10 Hz IF bandwidth. Since we measured with a broader 1 KHz IF bandwidth we added 20 dB to obtain the values in Table 4. We like to mention here that the values associated with $\langle E_{cr}^g \rangle$ for X-band and the hv channel of C-band were actually lower than the -120 dBm levels associated with $E_b$. We do not have a clear explanation for this. We therefore consider the $E_b$ as the absolute minimum signal levels and therefore adjusted the values to this level.

The variation of $\sigma_{min}$ over the bands and polarization channels is due to the variation in measured values of $E_0^{g0}$. Overall the minimum RCS is about -50 $\mathrm{m^2}$ (dB). Other studies use the more appropriate so-called noise-equivalent $\sigma^0$ ($\mathrm{m^2\,m^{-2}}$) to quantify the minimum detectable (distributed) target, see for example Nandan et al. (2016) or Nagarajan et al. (2014). Because

**Table 4.** Summary of systematic uncertainties, -offsets and minimum signal levels. Concerning $\Delta E_T$, $E_{cr}^g$, and $E_b$: table values are receiver power levels derived from measured S-parameters which, in their turn, are associated with $\Delta E_T$, $E_{cr}^g$, and $E_b$. With $\Delta K$ and $\sigma_{min}$ actual values are shown.

| | | L-band | S-band | C-band | X-band |
|---|---|---|---|---|---|
| Uncertainties | | | | | |
| **Temperature-induced radar return uncertainty** $\Delta E_T$. | vv | -95 | -98 | -95 | -103 |
| $dB(\Delta S_T) - 16 \text{ dB} + 10 \text{ dBm} = \text{(in dBm)} \rightarrow$ | vh | -107 | -103 | -103 | -104 |
| where $\Delta S_T$ is measured S-parameter associated | hv | -103 | -104 | -104 | -103 |
| with $\Delta E_T$. | hh | -98 | -92 | -96 | -103 |
| | | | | | |
| **Reference target measurement uncertainty** $\Delta K$. | vv | $\pm$ 0.1 | $\pm$ 0.1 | $\pm$ 0.2 | $\pm$ 1.0 |
| Relative error (in dB) $\rightarrow$ | vh | $\pm$ 0.4 | $\pm$ 0.1 | $\pm$ 0.2 | $\pm$ 0.8 |
| | hv | $\pm$ 0.4 | $\pm$ 0.1 | $\pm$ 0.2 | $\pm$ 0.8 |
| | hh | $\pm$ 0.1 | $\pm$ 0.1 | $\pm$ 0.3 | $\pm$ 1.0 |
| Offsets and minimum signal levels | | | | | |
| **Offset due to antenna coupling remnant** $E_{rc}^g$. | vv | -86 | -103 | -113 | -120 |
| $dB(\langle S_{cr}^g \rangle) - 16 \text{ dB} + 10 \text{ dBm} = \text{(in dBm)} \rightarrow$ | vh | -92 | -102 | -119 | -120 |
| where $\langle S_{cr}^g \rangle$ is measured S-parameter, averaged | hv | -96 | -104 | -120 | -120 |
| over $BW$, associated with $E_{cr}^g$. | hh | -82 | - 91 | -107 | -120 |
| | | | | | |
| **Minimum detectable signal level** $E_b$. | | | | | |
| $dB(S_b) - 16 \text{ dB} + 10 \text{ dBm} = \text{(in dBm)} \rightarrow$ | | -119 | -120 | -120 | -120 |
| where $S_b$ is measured S-parameter, averaged | | | | | |
| over $BW$, associated with $E_b$. | | | | | |
| | | | | | |
| **Minimum detectable RCS value** $\sigma_{min}$. | vv | -53 | -52 | -51 | -48 |
| Given target distance is $R_{fp}$ (m$^2$ expressed in dB) $\rightarrow$ | vh | -49 | -51 | -51 | -49 |
| | hv | -50 | -52 | -51 | -51 |
| | hh | -53 | -54 | -52 | -50 |

of our broad antenna radiation patterns, however, this quantity is not suitable and therefore we instead refer to a discrete target extending a small solid angle.

Starting with Eq. 6 it can be shown (see Appendix E4) that the three estimated types of uncertainty, namely fading, temperature-induced radar return uncertainty ($\Delta E_T$), and reference target measurement uncertainty ($\Delta K$) can be combined in

a model for total $\sigma^0$ uncertainty:

$$\sigma^0 = \frac{I_N \pm \Delta I_N}{(K \pm \frac{2}{3}\Delta K)(1 \pm 1/\sqrt{N})} = \frac{I_N}{K} \pm \Delta\sigma^0 \tag{12}$$

$\Delta I_N$ (W m$^{-2}$) is a statistical error that follows from $\Delta E_T$, $\Delta K$ is converted from a maximum possible error into a statistical error with a (2/3) probability confidence interval and the term $1/\sqrt{N}$ represents a statistical error caused by fading. In the right term the three uncertainty contributions are merged into one statistical uncertainty $\Delta\sigma^0$ (m$^2$ m$^{-2}$), which is a 66% confidence interval for $\sigma_0$. In this paper these 66% confidence intervals are presented in all figures showing our retrieved $\sigma^0$. To give an indication of the magnitude of $\Delta\sigma^0$, which are different per bandwidth, polarization, and overall $\sigma^0$-level, some extremes are summarized in Table 5. Shown values were retrieved from the calculated time-series results, which are presented in Section 5.2.3.

**Table 5.** Example uncertainty values $\Delta\sigma^0$ (dB) per bandwidth, polarization, and overall $\sigma^0$-level.

| | L-band | S-band | C-band | X-band |
|---|---|---|---|---|
| High $\sigma^0$ -levels (typical in summer) | | | | |
| vv | +1.6 – -2.5 | +1.3 – -1.9 | +1.4 – -2.1 | +1.7 – -3.0 |
| vh | +1.7 – -3.0 | +1.3 – -1.9 | +1.4 – -2.2 | +1.6 – -2.7 |
| hv | +1.8 – -3.2 | +1.3 – -1.9 | +1.4 – -2.0 | +1.6 – -2.7 |
| hh | +1.6 – -2.5 | +1.2 – -1.7 | +1.3 – -2.0 | +1.7 – -2.9 |
| Low $\sigma^0$ -levels (typical in winter) | | | | |
| vv | +2.3 – -5.2 | +1.9 – -3.7 | +1.7 – -2.9 | +2.1 – -4.2 |
| vh | +2.3 – -5.2 | +2.4 – -5.9 | +2.6 – -8.3 | +2.3 – -5.2 |
| hv | +2.4 – -6.0 | +2.5 – -6.6 | +2.5 – -6.4 | +2.0 – -4.9 |
| hh | +2.3 – -5.3 | +1.7 – -2.8 | +1.7 – -2.7 | +1.9 – -3.8 |

### 5.1.2 Uncertainty due to angular resolution antenna patterns

Measuring the dependence of $\sigma^0$ on incidence angle $\theta$, $\sigma^0(\theta)$, with a scatterometer whose antenna radiation patterns are $G(\alpha, \beta)$ is equivalent to the convolution of $\sigma^0(\theta)$ with $G(\alpha[\theta], \beta[\theta])$. For a narrow-beamwidth antenna $G(\alpha[\theta], \beta[\theta])$ may be approximated by a block-function whose width is the FWHM beamwidth. This is equivalent to the narrow-beam approximation mentioned in Sec. 4.1, the measured 'convolved' $\sigma^0(\theta)$ is similar to the 'actual' $\sigma^0(\theta)$. With antennas whose FWHM beamwidths probably exceed the rate of change of $\sigma^0$ over $\theta$ this approximation will lead to larger errors. Still, in principle it is possible to deconvolve the convoluted $\sigma^0(\theta)$ function to obtain the actual $\sigma^0(\theta)$ since $G(\alpha, \beta)$ is known. This deconvolution is performed by Axline (1974) for example, but was considered to be outside the scope of this paper. Instead, the procedure as explained in Sec. 4.1 was followed which, consequently, does result in an unknown uncertainty in the retrieved $\sigma^0$.

It is possible, however, to estimate this uncertainty with a simple numerical experiment in which the scatterometer return is simulated using a pre-defined functional type of $\sigma^0(\theta)$. We used the empirical model $\sigma^0_{pq}(\theta)$ for grassland developed by Ulaby

and Dobson (1989). When using our retreival method on the simulated scatterometer return we obtain, for 4.75 GHz with vv polarization $\sigma_{vv}^0 = -14.4$ dB for $34° \leq \theta \leq 60°$, while the actual value over this interval varies from $-13.0 \leq \sigma_{vv}^0 \leq -14.9$ dB. Although this discrepancy depends on the (unknown) form of $\sigma^0(\theta)$, in general this error will be larger for low- and smaller for high frequencies because of the respective antenna beamwidths.

## 5.2 Measured backscattering coefficients

For the remaining analysis we discuss results of four bandwidths $BW$, picked amidst frequency ranges typically used in microwave remote sensing: 9 – 10 GHz (X-band), 4.5 – 5 GHz (C-band), 2.5 – 3 GHz (S-band), and 1.5 – 1.75 GHz (L-band). The widths decrease with wavelength due to the expected frequency resolution of the target's scattering response (Sec. 4.3) and the antenna-radiation-pattern change over frequency (Appendix D).

### 5.2.1 Angular variation of $\sigma_{pp}^0$ for asphalt

We start with the asphalt experiment results, which we present here to demonstrate that our $\sigma^0$ retrieval method, using measurement data obtained with our scatterometer system, results in $\sigma^0$ values comparable to those in other studies.

Figure 9 shows our retrieved $\sigma_{pq}^0$ over $\alpha_0$ for all bandwidths and polarization channels. Since with all bands the uncertainty intervals for vh and hv overlap we only show vh cross polarization channel for figure clarity. When comparing the results for S-, C-, and X-band we observe an increase in backscatter over frequency, which can be explained by the increment of the surface roughness to wavelength ratio. For X-, and C-band the vv backscatter is stronger than with hh. For S-band this also holds, although the comparison is more difficult as the $\theta$ intervals become broader. It is clear however, that for all bands the cross-response is lower than that of the co polarization. Remarkable, at first sight, is that the retrieved $\sigma^0$ for L-band is higher than that of S-band. We believe this is due to the lowest angular resolution of our system at L-band and our subsequent $\sigma^0$ retrieval method from the measured signal. As shown in the graphs, for L-band the backscatter from near-nadir $\theta$ angles are included in the received signal for almost all $\alpha_0$ angular positions. As the 'actual' $\sigma^0(\theta)$, in general, shoots upward for the smaller $\theta$-angles towards the peak value at nadir the resulting signal, and with it, the retrieved $\sigma^0$ is high as well.

Our results are plotted together with those found in other studies. Baldi (2014) also measured asphalt backscatter for S-band. His scatterometer had a more narrow beamwidth of $10°$, allowing for a straightforward measurement of $\sigma^0$ over $\theta$. He measured over $15° \leq \theta \leq 55°$. For a comparison to our results, we used his measured $\sigma^0(\theta)$ in Eq. 4 and subsequently applied our retrieval method to this simulated radar return $P^{Rx}$. The resulting $\sigma^0$ values are shown in 9. Three points for vv-, and two for vh polarization could be retrieved. Because no data was presented outside the $15° - 55°$-range the hh polarization response could not be simulated. In general, we consider our results to match with Baldi's satisfactory. The differences may be attributed to fading uncertainty (low number of spatial samples) and to different surface roughness values: it seems our asphalt was smoother. However, the latter argument is speculative since neither we nor Baldi measured the surface roughness.

The only other study on L-band backscatter from asphalt we could find was that by Peake and Oliver (1971). There $\sigma^0$ values are reported for smooth asphalt with an estimated surface roughness of $s = 0.3$ mm for $20° \leq \theta \leq 70°$ for vv and $10° \leq \theta \leq 70°$ for hh. Because of the broad L-band $\theta$-ranges for our scatterometer, however, a simulation of the $\sigma^0$-retrieval, as with Baldi's data, would be incorrect.

For X-band with co-polarization we compare our results with the empirical model for asphalt described in Ulaby and Dobson (1989). This model is formed using measurements from multiple other studies with asphalt having various roughness values. Since our antenna beamwidths at X-band are sufficiently narrow we can compare our results without further adjustment. No empirical model is given for asphalt at X-band with cross polarization in Ulaby and Dobson (1989). For both vv- and hh polarization our retrieved $\sigma^0$ shows a clear overall decreasing trend over $\theta$, which is expected for a surface that is smooth compared to the wavelength. Overall, $\sigma^0$ for vv polarization is higher than for hh polarization, which is in accordance to the empirical model. Starting from the smaller angles, the consecutive measurement points remain at similar level. With hh polarization there appears to be even a local minimum at $40°$, although the measurement uncertainty is relatively large there. Given that the empirical curves show a similar trend, though not as pronounced, the slow decay of $\sigma^0_{pp}$ over $\theta$ for $25 - 55°$ can simply be a property of asphalt. Overall we find our measurements to lie within the 90 % occurrence interval of the empirical model and therefore conclude that our results for asphalt are similar to those of Ulaby and Dobson (1989). We could not find studies reporting asphalt backscatter for C-band.

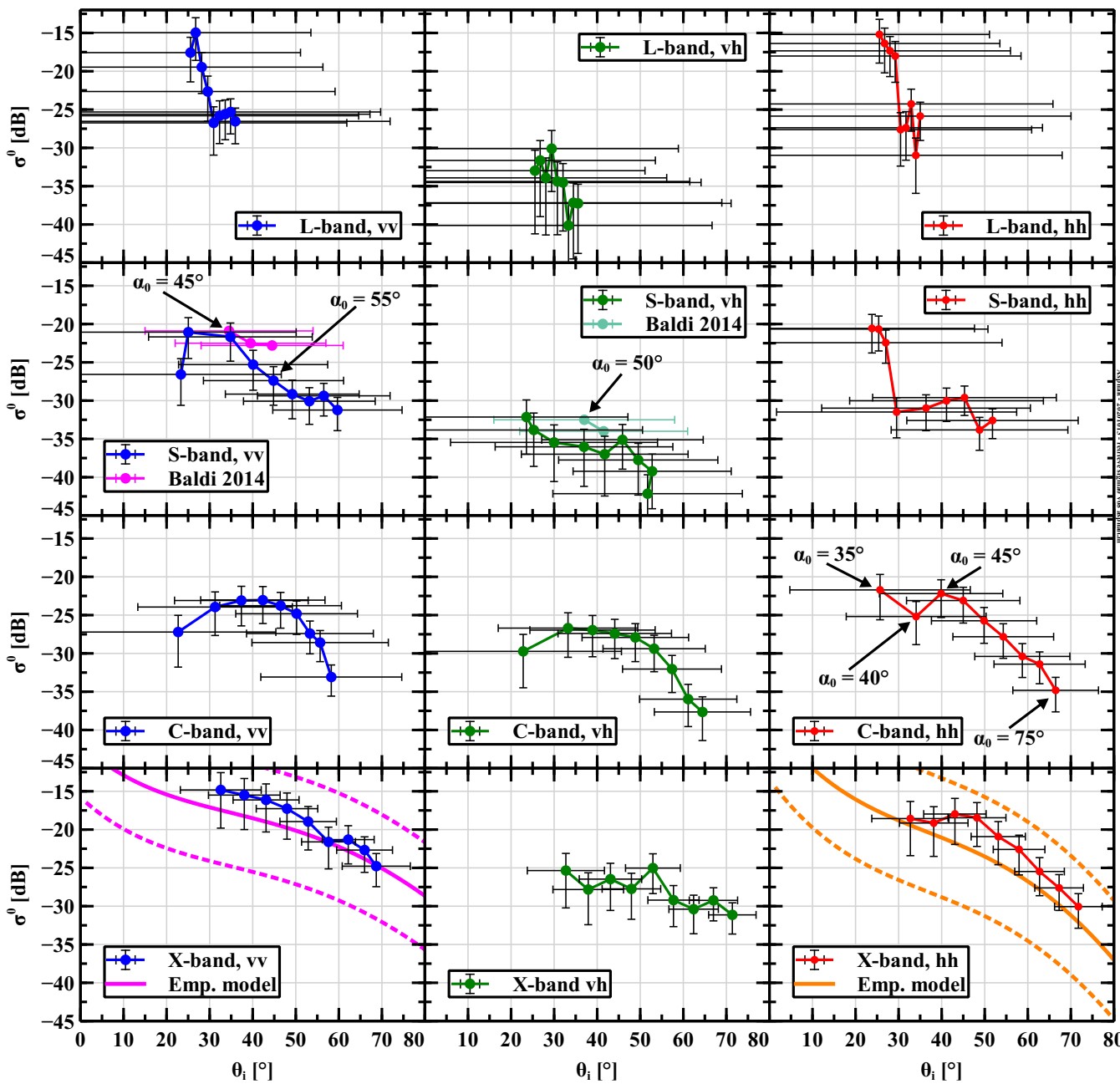

**Figure 9.** Measurement results of $\sigma_{pq}^0(\alpha_0)$ for all bands and polarizations together with S-band measurement results from Baldi (2014) and empirical model for X-band from Ulaby and Dobson (1989). Points represent results for different antenna boresight angles $\alpha_0$. Horizontal bars represent intervals for angle of incidence $\theta$ and vertical bars the 66% confidence interval for $\sigma^0$. Dotted lines between data points are guide to the eye. With X-band, solid and dotted curves (magenta and orange) represent mean value and 90% confidence interval of empirical model respectively.

### 5.2.2 Angular variation of $\sigma^0_{pq}$ in Maqu

We present next the measurement results and analysis of the angle-dependent backscatter of the Maqu-site surface with the following purposes. First, to quantify the behaviour of $\sigma^0$ with respect to the elevation angle ($\theta$), $BW$, and polarization channels for the Maqu site ground surface with a living vegetation canopy. Second, to assess the spatial homogeneity of $\sigma^0(\theta)$ over the Maqu-site surface by also measuring backscatter at different azimuth angles ($\phi$). As explained in Sec. 4.3, the single footprint area for the $\sigma^0$ time-series measurements should be representative for the whole Maqu-site surface.

Due to practical limitations of possible $\phi$ angles and because of the wide antenna beam widths, the footprints of used $\alpha_0$- and $\phi$ combinations in this experiment overlap partially, as is shown in Fig. 3. However, since we employ frequency averaging to reduce the fading uncertainty for every footprint, we argue that the $\sigma^0$ -values retrieved per (overlapping) footprint may nevertheless be compared to each other for this section's analysis.

Figure 10 shows, as example, retrieved backscattering coefficients for different $\alpha_0$- and $\phi$ angles for all bandwidths at one polarization channel. There is a clear tendency of $\sigma_0$ decreasing over $\alpha_0$, as it should. Deviations from this trend, for example with X-band at $\phi = 10°$, $\alpha_0 = 50°$, might point to local strong scattering, but could also simply be due to fading. From the analysis that follows we conclude in favour of the latter.

As a means to quantitatively evaluate the $\sigma^0$ behaviour with respect to $\theta$- and $\phi$ angle the data is grouped in sets of $\sigma^0$ over $\alpha_0$ for every angle $\phi$, $BW$, and polarization. Next, an iterative least-squares non-linear fitting algorithm is applied to fit each set to the model

$$\sigma^0 = A\cos(\theta)^B \tag{13}$$

where $A$ is a constant ($\mathrm{m^2\,m^{-2}}$) and $B$ is either 1 for an isotropic scatterer or 2 for a surface in accordance with Lambert's law (Clapp, 1946). For each $\alpha_0$ we find the coordinate for which $G^2/R^4$ is maximum and use that position's angle of incidence $\theta$ together with the centre $\sigma^0$ -value of the 66% confidence interval for the fitting process. As a next step, we reduced the number of fitting possibilities by selecting for each polarization-$BW$ combination the most likely value for $B$ (1 or 2). This was done by tallying over the $\phi$ -angles which of the two fitted curves $\sigma^0 = A\cos(\theta)^B$ passed through the confidence intervals best and had the highest coefficients of determination ($R^2$). The outcome was $B = 1$ for all polarization channels of X-band and $B = 2$ for all of S-, and L-band. For C-band it was harder to judge in favour of either. We chose $B = 1$ for vh polarization and $B = 2$ for vv, hh, and hv. An overview for found parameters $A$ and $B$ is presented in Fig. 11. The left column shows the best results, i.e. having the favourable $B$ -value, for all bandwidths and polarizations. The numbers next to all $A$ -values represent their $R^2$ -values.

We comment first on the found $B$ coefficients which characterize the angular dependence $\sigma^0(\theta)$. The stronger decrease over angle found with L- and S-band is as expected since for longer wavelengths there is less volume scattering from the vegetation canopy and the soil reflections become more dominant. For these longer wavelengths the soil surface roughness appears smoother, causing specular reflection to be stronger and non-specular reflections (including in the backward direction)

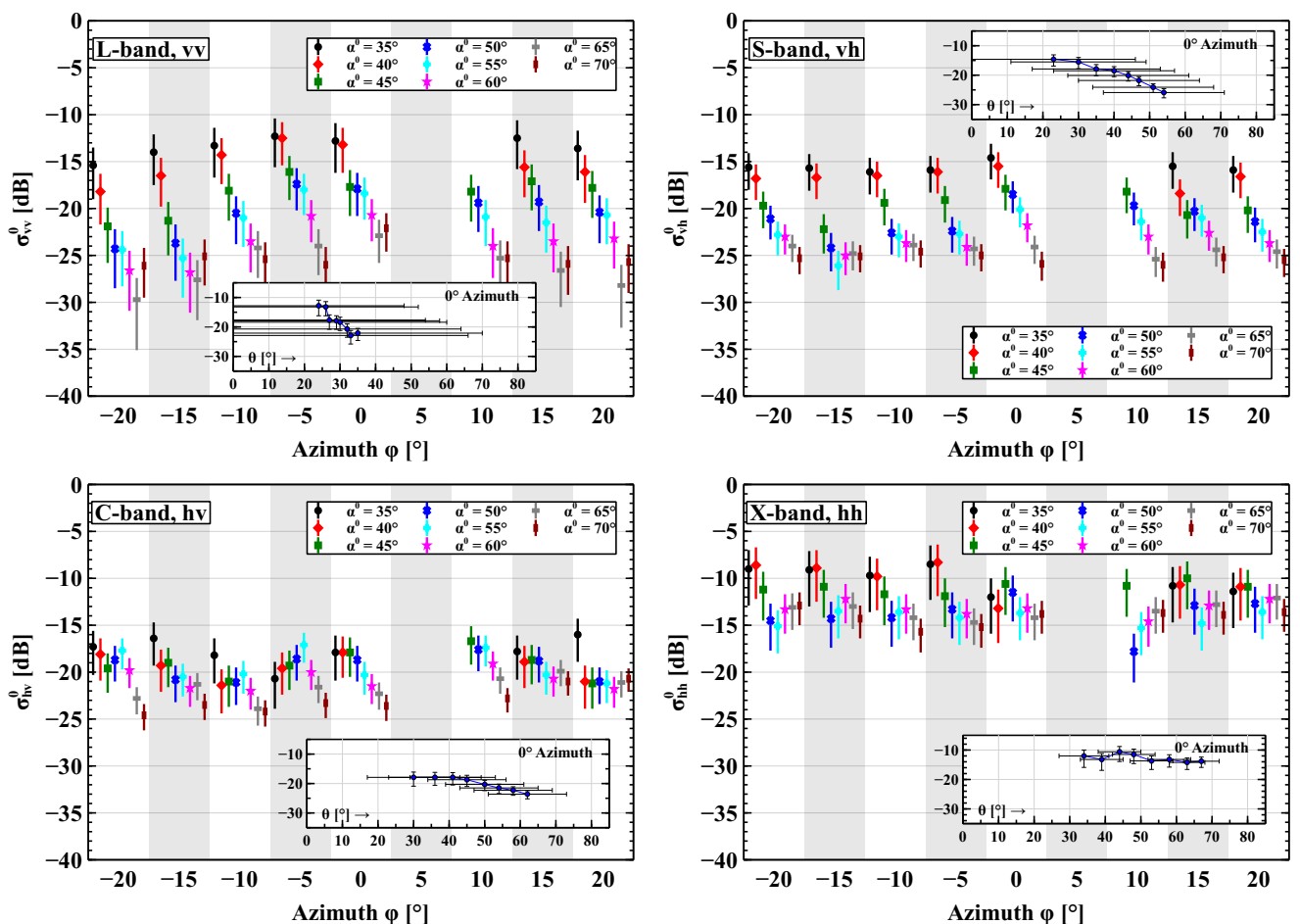

**Figure 10.** Measurement of $\sigma_{pq}^0(\alpha_0, \phi)$ for all bandwidths at different polarization over the Maqu site on 2017 08 25. Four main figures: For different antenna boresight azimuth angles $\phi$ the variation of $\sigma_{pq}^0$ over boresight elevation angles $\alpha_0$ is shown. The eight vertical bars represent the 66% confidence interval for $\sigma^0$. Intervals for incidence angles $\theta$ per measurement are not shown here for clarity of figure. Insets: $\sigma_{pq}^0(\alpha_0)$ for $\phi = 0°$. Horizontal bars represent intervals of actual incidence angles $\theta$, which are identical for other $\phi$-values in main figures.

to decrease more rapidly with $\theta$. This effect is well-known, see for example (de Roo and Ulaby, 1994). By the same logic, for X-band $\sigma^0$ will decrease more slowly over $\theta$ as scattering from the vegetation canopy becomes dominant over that from the soil surface. Strong vegetation scattering is known to be more constant over $\theta$ (see for example Stiles et al. (2000)) and thus the model for an isotropic scattering surface, i.e. $B = 1$ is more suitable. With C-band both $B = 1$ and $B = 2$ fitted best for about half of the $\phi$ angles which indicates that at this 'intermediate' wavelength we see both aforementioned features.

Next we focus on the found magnitudes of $A$. With the co-polarization channels we see that, on average over $\phi$, the amount of backscatter decreases with increasing wavelength as expected considering the description above. An exception however,

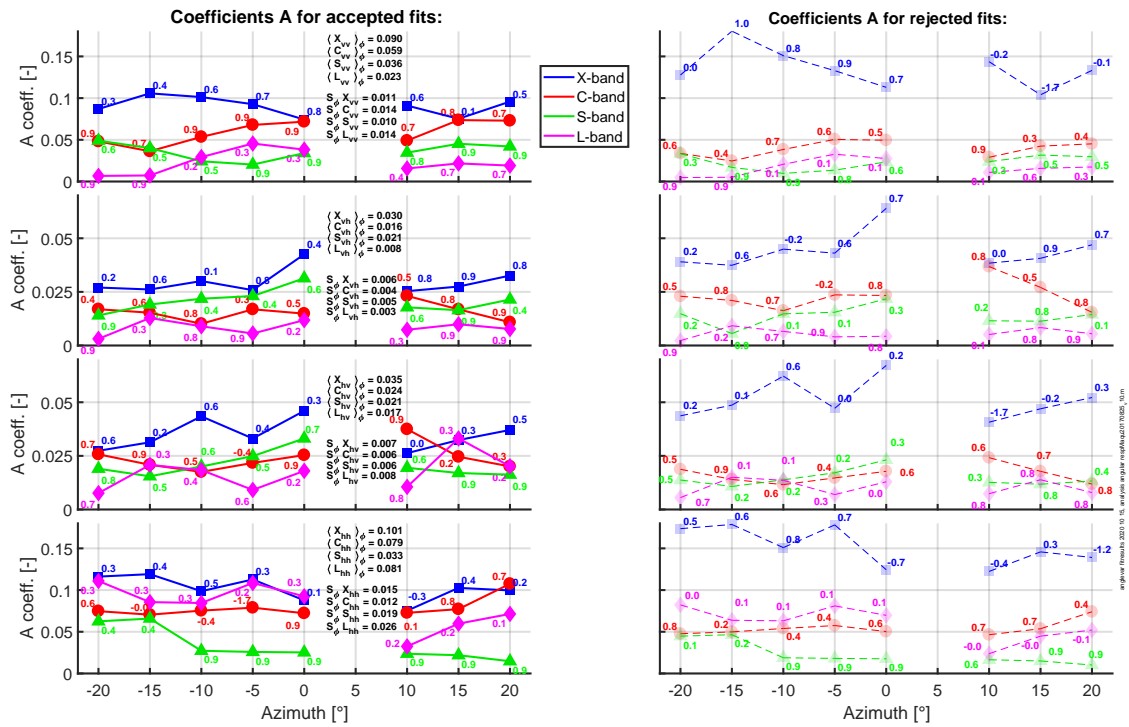

**Figure 11.** Results of fitting the derived values $\sigma^0_{pq}$ over $\alpha_0$ to model $\sigma_0(\theta) = Acos(\theta)^B$ for different azimuth angles $\phi$, bandwidths $BW$, and polarization channels. Left column shows found coefficients $A$ over $\phi$ for best fits with favourable $B$-value for each $BW$ and polarization and right column the $A$ coefficients with less favourable $B$-values. Numbers at data points indicate coefficient of determination ($R^2$) of individual fits. Values in the centre are average $\langle B_{pq}\rangle_\phi$ and standard deviation $S_\phi B_{pq}$ over $\phi$, with $B = L, S, C,$ or $X$ as bandwidth.

is the L-band response with hh polarization which is comparable to that of C-band. As with the asphalt measurements (sec. 5.2.1), we believe these high $\sigma^0$ retrievals are due to the low angular resolution of our scatterometer for L-band. As a result, the
535 backscatter for close to nadir angles (which are in general the highest) is present in all angular positions $\alpha_0$. This is visible in the inset figure of Fig. 10. We also note that the variation over $\phi$ (by comparing $S_\phi B_{pp}$ to $\langle B_{pq}\rangle_\phi$) is smallest for X-, and largest for L-band. The cross polarization response is lower than that for co as expected. For both vh and hv the X-band backscatter is also largest here while those for L-band are lowest. However, S-band appears to have stronger backscatter than C-band. We do not have a clear explanation for this. As with the co polarization channels the variation over $\phi$ is strongest for the longer
wavelengths.

Finally some remarks on the variation of $A$ over $\phi$ and, virtually, arccos the surface area. Except for X-band with hh polarizations there did not appear to be a systematic trend of $A$ over $\phi$. Also, there was not one particular $\phi$ angle for which the values for $A$ over $BW$ and polarization stood out from the rest. These observations indicate that the surface area covered by our scatterometer appeared to have uniform (scattering) properties. The somewhat higher $A$ values with the negative $\phi$ values with X-band at hh polarization are probably caused by a difference in vegetation density between the left- and right side of the Maqu site. Fortunately, for $\phi = 0°$ the $A$ value had a medium value compared to the other $\phi$ angles, so that we may still interpret the surface area associated with the scatterometer's (fixed) footprint during the time-series measurements as being representative for its surroundings.

### 5.2.3   Time-series of $\sigma^0_{pq}$ in Maqu

Presented in this section is first, a global overview of the retrieved $\sigma^0_{pq}$ over the period 26 August 2017 – 26 August 2018. We then present graphs showing $\sigma^0_{pq}$ over four different 13-day periods at the highest temporal resolution. Of these we shall briefly discuss the third feature, which shows the $\sigma^0_{pq}$- and $M_v$ dynamics during the thawing period at the beginning of April 2018.

Figure 12 presents an overview of the time-series data of $\sigma^0_{pq}$ over the whole August 2017 – 2018 period for all considered bandwidths in L-, S-, C-, and X-band, along with $M_v$ and $T_{soil}$ at four depths ranging from 2.5 to 20 cm. For visibility reasons the graphs only display measurements taken at 18:10 with 2 day intervals. Data of the radar return and $\sigma^0_{pp}$ for November 2017 is not available, while that of late June – Early July 2018 will become available at a later stage.

We observe for all bands and polarizations that $\sigma^0$ is highest in summer and autumn while it is lowest during winter. This may be explained by the fact that in summer and autumn $M_v$, and the amount of fresh biomass is highest. As a result, the high dielectric constant of moist soil, in combination with the rough surface and presence of water in the vegetation results in strong backscattering. During winter, however, there is little liquid water, i.e. $M_v$, present in the soil and no fresh biomass (dry biomass however remains present, see Fig. A6). The dielectric constant of the soil therefore is lower compared to that of moist soil and there is little to no scattering from the dried out vegetation, resulting in a lower $\sigma^0_{pq}$. There were however peaks of $\sigma^0_{pq}$ during winter, for example on 26 January, which coincided with snowfall. Snow cover, deposited on the layer of dead vegetation, forms a rough surface that allows for strong backscatter. The dynamics of $\sigma^0_{pq}$ during thawing period will be discussed in more detail below.

When comparing the four bands we observe that, in general, the backscattering is highest for X-band and lowest for L-band or S-band. This difference is mainly driven by the wavelength-dependent response to the surface roughness of the soil and vegetation during the summer and autumn period. For longer wavelengths the soil surface roughness appears smoother than for the shorter wavelengths, resulting in stronger specular reflection, thus lower backscatter. A similar argument holds for the vegetation; its constituents are small compared to the longer wavelengths, thus little volume scattering occurs. An important reason that the retrieved $\sigma^0_{pq}$ for L-band is similar to, or sometimes even greater than S-band is because of the low angular resolution at this wavelength as pointed out in the previous sections. Since the backscatter for near-nadir $\theta$-angles, which

is highest in general, is included in the L-band radar return the retrieved $\sigma^0_{pq}$ is higher then one would expect based on the wavelength-dependent scattering properties of the ground alone. However, there are also plausible target-associated mechanism which increase backscatter particularly for longer wavelengths. During the freezing period top soil freezes first while the deeper layers still contain high(er) amounts of liquid water. The penetration depth for shorter wavelengths may be insufficient for reaching these, strongly scattering, layers but sufficient for the longer wavelengths, resulting in a stronger radar return. Also, the hanging long grass in summer, combined with the low angular-resolution effect, can give rise to a high radar return and, subsequently a high retrieved $\sigma^0$ for hh polarization for L-band. This can be seen in Fig. 16) where the L-band $\sigma^0_{hh}$ exceeds that of S-band by about 4 dB.

Except for during the summer, backscatter for vv polarization was equal to, or higher than that for hh polarization. This behaviour was also observed by Oh et al. (1992), albeit for bare soil. We however may compare our situation to that of bare soil during winter, when there is no fresh biomass. When vegetation was present, $\sigma^0_{hh}$ was stronger for all bands, as is visible during June - August 2018. This was however not the case during August - September 2017, when the vegetation probably still contained water. Somewhat stronger backscatter, $0.5 - 1$ dB, for hh- than for vv polarization was also reported for grassland in Ulaby and Dobson (1989) with $40 \leq \theta \leq 60°$ for S- and X-band. For C-band they reported no clear difference. Yet another study, (Kim et al., 2014), measured 3-4 dB higher backscatter for hh- than for vv polarization when measuring wheat at L-band ($\theta = 40°$). Our results for L-band were similar. Cross polarization $\sigma^0$ levels were, as expected, lower than those of co polarization. During winter period this difference was largest, especially with C-band. For L-band, on the other hand, this difference in $\sigma^0$ levels between co- and cross polarization was quite small.

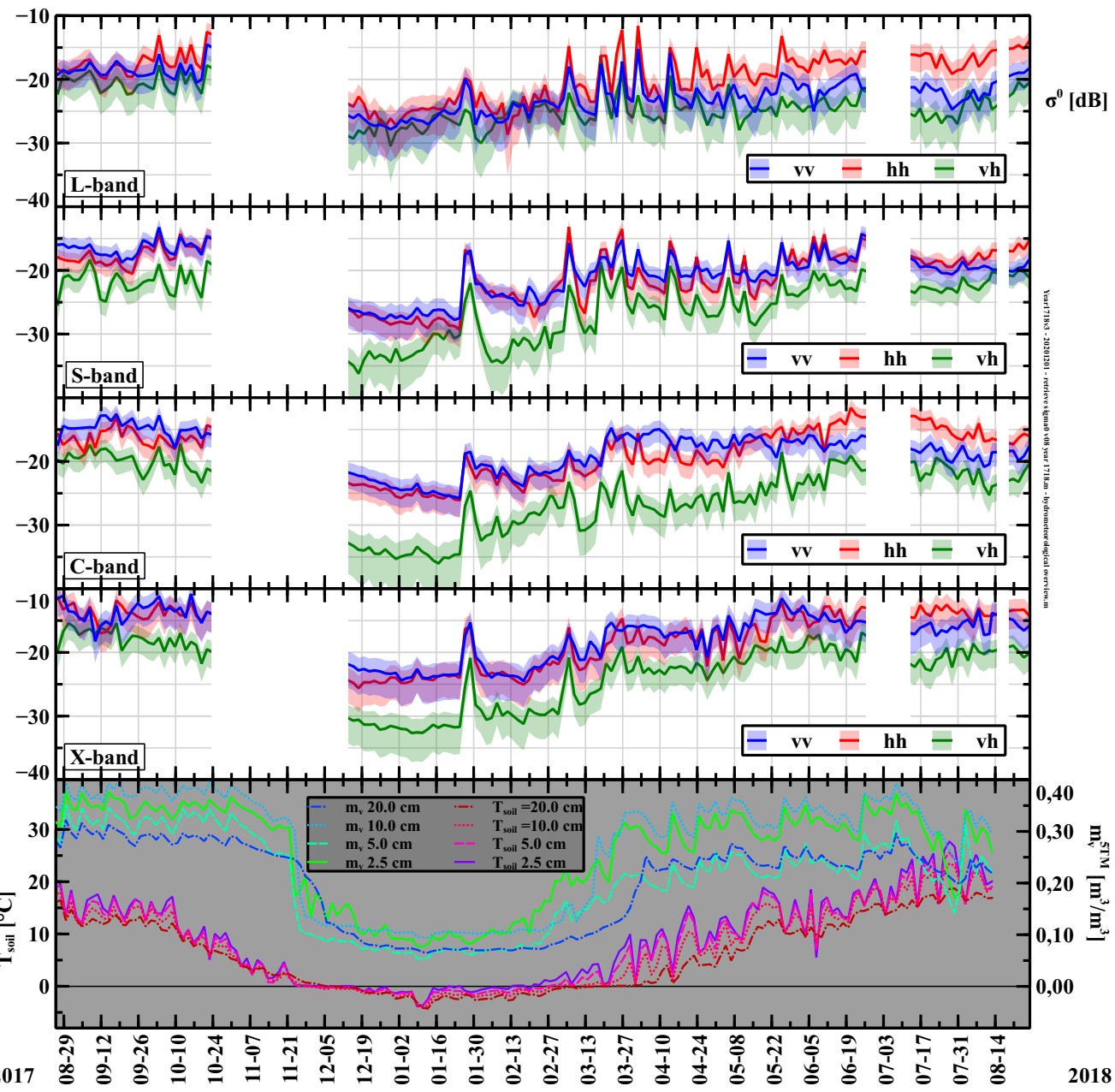

**Figure 12.** Time-series measurements of $\sigma_{pq}^0$ ($\mathrm{m^2\,m^{-2}}$) for L-, S-, C- and X-band, $M_v$ and $T_{soil}$ from August 2017 to 2018. Shown are measurements taken at 18:10 with 2 day intervals. Shaded regions indicate 66% confidence intervals for $\sigma_{pq}^0$. Antenna boresight angle fixed at $\alpha_0 = 55°$. The incidence angle ranges are band- and polarization dependent. Widest ranges are: L-band: $0° \le \theta \le 60°$, S-band: $20° \le \theta \le 60°$, C-band: $36° \le \theta \le 60°$, and X-band: $47° \le \theta \le 59°$. Bottom graphs show measured volumetric soil moisture $m_v^{5TM}$ ($\mathrm{m^3\,m^{-3}}$) and soil temperature $T_{soil}$ at indicated depths. Spatial average volumetric soil moisture $M_v$ is estimated as $M_v = m_v^{5TM} \pm 0.04\ \mathrm{m^3\,m^{-3}}$.

For four 13-day periods the retrieved $\sigma^0$ at 30-minute intervals is shown in Figures 13 (October 2017), 14 (December 2017), 15 (April 2018), and 16 (July 2018). When selecting these periods we tried avoiding strong precipitation events as much as possible, since these complicate the interpretation. Besides $M_v$ and $T_{soil}$ at four depths the precipitation rate ($\mathrm{mm\,hr^{-1}}$) is shown as well.

We shall describe the retrieved $\sigma^0_{pq}$ during 13-day period in April 2018 (Fig. 15) when the thawing process has started. The most prominent features in the measured backscatter are the diurnal variations of $\sigma^0_{pq}$ that are clearly caused by changes of $M^v$. For S-, C-, and X- bands we observe that $\sigma^0$ increases during daytime due to the increase of liquid water in the top soil due to thawing and at night $\sigma^0$ drops as most of the water freezes again. For L-band this behaviour is also visible, though not as pronounced. The $M^v$ changes at different depths are consistent with this difference: the strongest diurnal variation in liquid water was measured by the probes at 2.5 and 5 $\mathrm{cm}$ depth while those at 10 and 20 $\mathrm{cm}$ do not change as much. With some days, for example on 4 and 5, or on 10 April, we observe diurnal changes in $\sigma^0$ (most pronounced for X-band) while the $M_v$ measured by the 5TM sensors at 2.5 and 5 $\mathrm{cm}$ depth showed little variations. This may suggest that the freezing and thawing during those days occurred only in the very top-soil layer, just below the air-soil interface where it was outside the influence zone of the 5TM sensors. The time lag between the drop of $\sigma^0$ (first) and the drop of 5TM $M_v$ (second), is caused by the same phenomena as the freezing starts at the top soil layer and progresses downward. The time lag during thawing was smaller. In general the magnitude of the $\sigma^0$ -change was largest for X-band and smallest for L-band, though exceptions exist. See for example 3 April, where for L-band $\sigma^0_{hh}$ drops almost 10 $\mathrm{dB}$, which is more than with the other bands. At the same time $M_v$ at 20 $\mathrm{cm}$ depth also shows strong variation, while $M_v$ at 10 $\mathrm{cm}$ changes less.

The presented 13-day periods alone already show there are a lot of interesting features contained in the backscatter signals. However, further investigation goes beyond the scope of this data paper. Our preliminary analysis demonstrates that the scatterometer data set collected at fixed time-intervals over a full year at the Maqu site contains valuable information on exchange of water and energy at the land-atmosphere interface. Information which is difficult to quantify with in-situ measurements techniques alone. Hence further investigation of this scatterometer data set provides an opportunity to gain new insights in hydro-meteorological processes, such as freezing and thawing, and how these can be monitored with multi-frequency scatterometer observations.

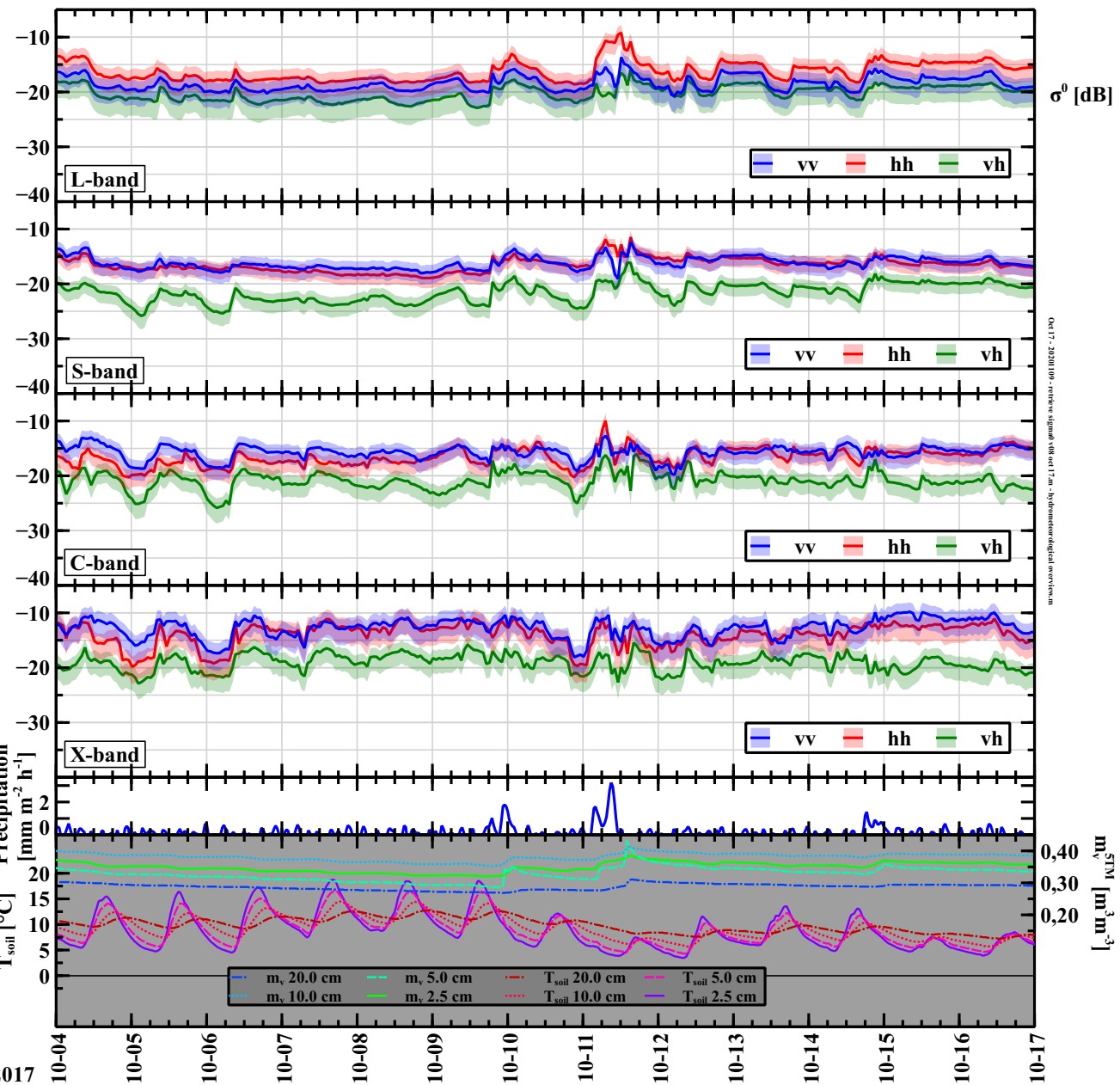

**Figure 13.** Time-series measurements of $\sigma_{pq}^0$ (m$^2$ m$^{-2}$) for L-, S-, C- and X-band, precipitation, $M_v$ and $T_{soil}$ during 13 days in October 2017. Shaded regions indicate 66% confidence intervals for $\sigma_{pq}^0$. Antenna boresight angle fixed at $\alpha_0 = 55°$. The incidence angle ranges are band- and polarization dependent. Widest ranges are: L-band: $0° \le \theta \le 60°$, S-band: $20° \le \theta \le 60°$, C-band: $36° \le \theta \le 60°$, and X-band: $47° \le \theta \le 59°$. Bottom graphs show measured precipitation (mm hr$^{-1}$), volumetric soil moisture $m_v^{5TM}$ (m$^3$ m$^{-3}$), and soil temperature $T_{soil}$ at indicated depths. Spatial average volumetric soil moisture content $M_v$ is estimated as $M_v = m_v^{5TM} \pm 0.04$ m$^3$ m$^{-3}$.

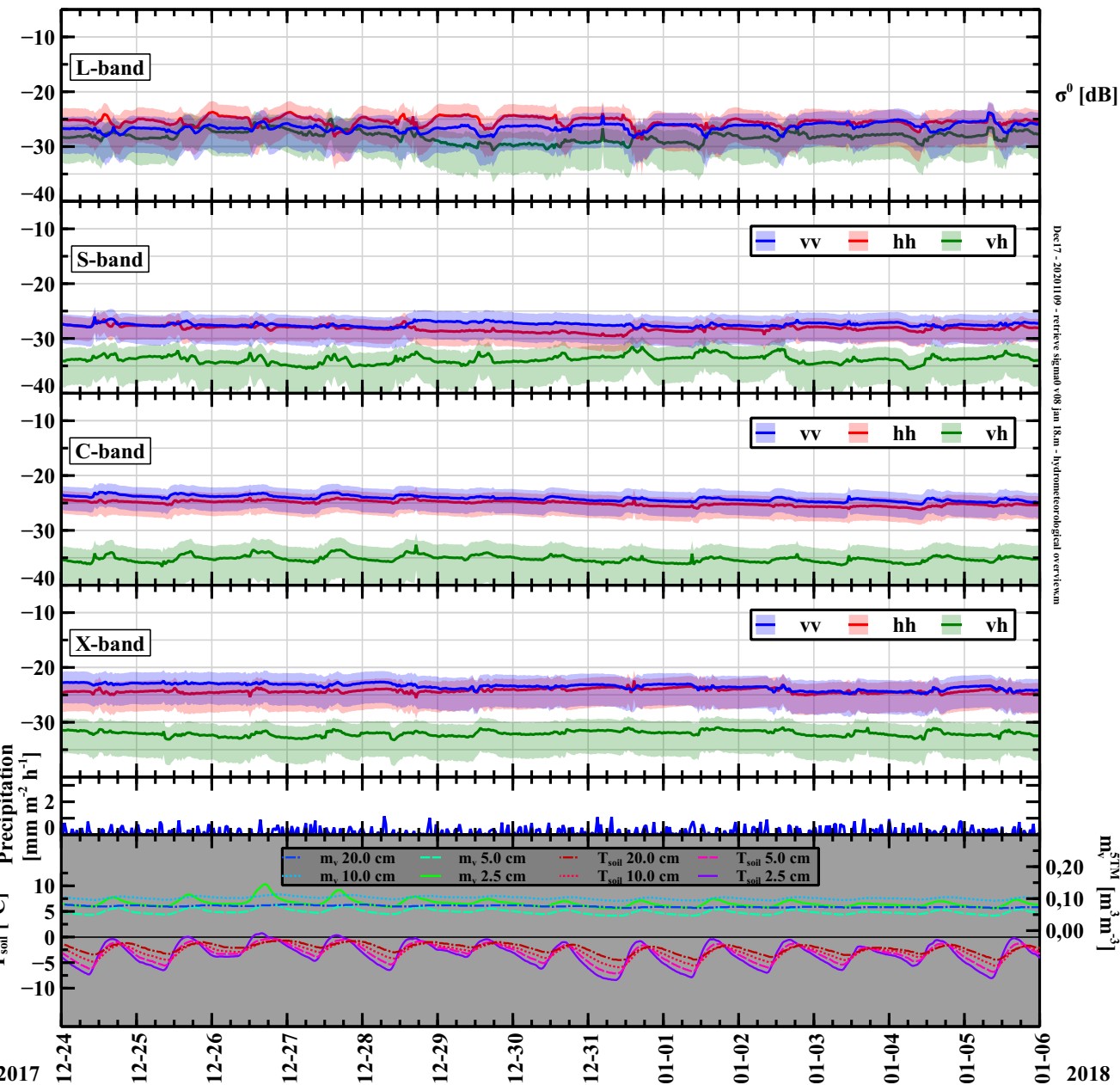

**Figure 14.** Time-series measurements of $\sigma^0_{pq}$ (m$^2$ m$^{-2}$) for L-, S-, C- and X-band, precipitation, $M_v$ and $T_{soil}$ during 13 days in December 2017. Same configurations as Fig. 13 apply.

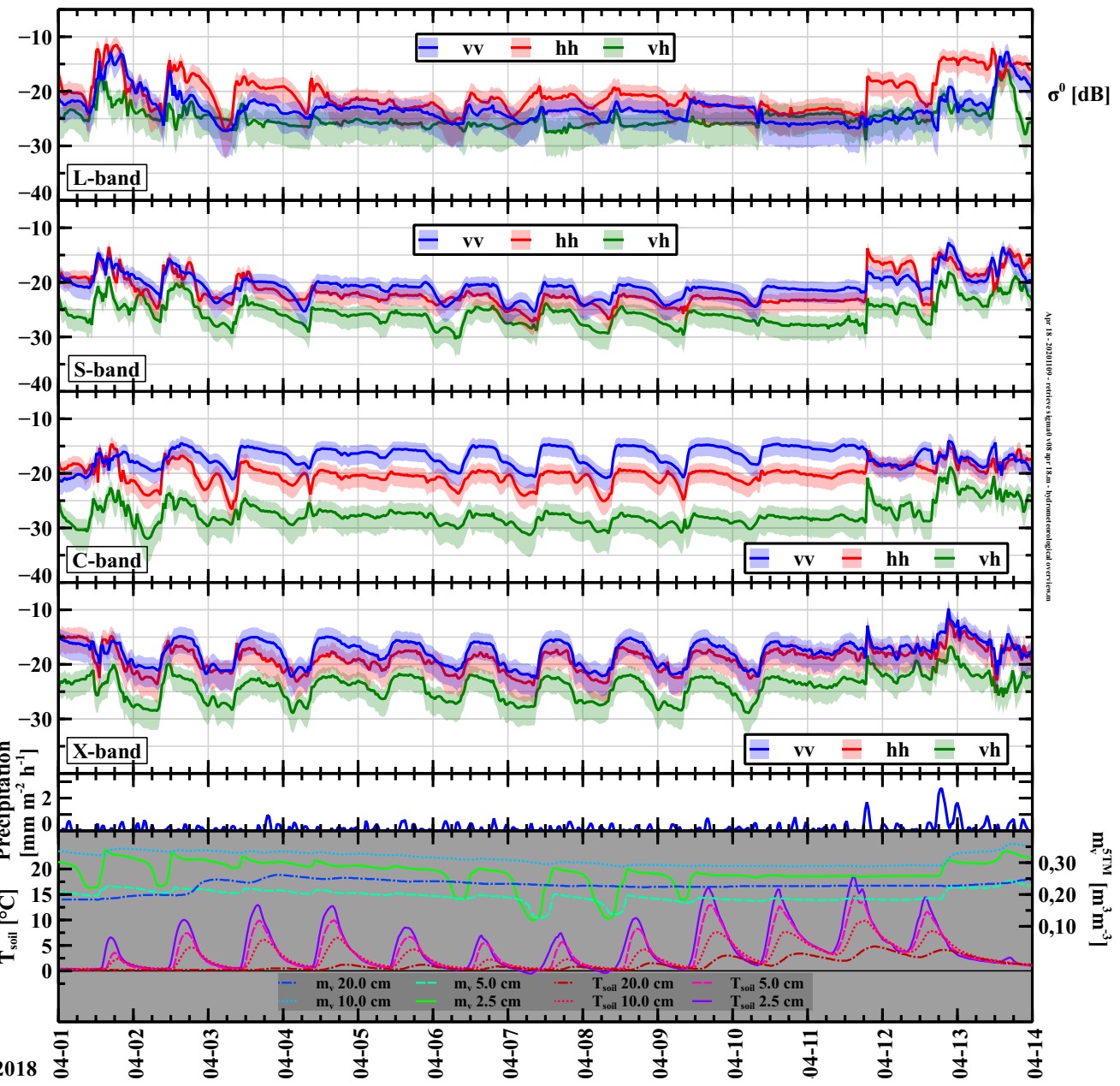

**Figure 15.** Time-series measurements of $\sigma_{pq}^0$ (m$^2$ m$^{-2}$) for L-, S-, C- and X-band, precipitation, $M_v$ and $T_{soil}$ during 13 days in April 2018. Same configurations as Fig. 13 apply.

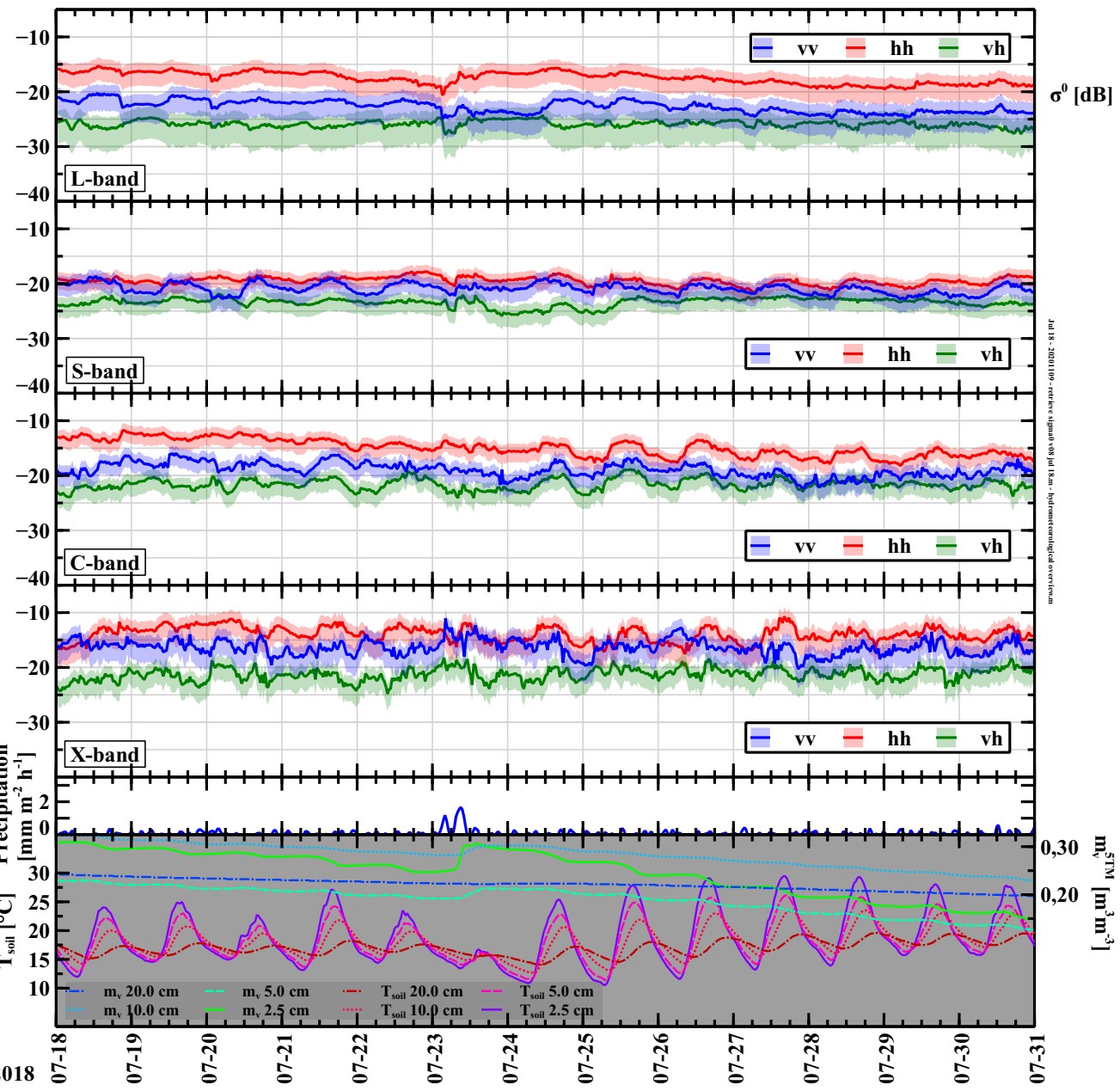

**Figure 16.** Time-series measurements of $\sigma_{pq}^0$ ($\mathrm{m^2\,m^{-2}}$) for L-, S-, C- and X-band, precipitation, $M_v$ and $T_{soil}$ during 13 days in July 2018. Same configurations as Fig. 13 apply.

## 6 Data Availability

In the DANS repository, under the link https://doi.org/10.17026/dans-zjk-rzts the collected scatterometer data is publicly available (Hofste et al., 2020). Stored are both the radar-return amplitude and phase for all four linear polarization combinations and processed $\sigma_{pq}^0$ for the L-, S-, C-, and X-band bandwidths discussed in this paper. The dataset includes time-series measurements from 26 August 2017 – 26 August 2018, data of angular variation experiments, and radar returns of the reference targets. Accompanying data includes time-series measurements of soil moisture and -temperature profile at depths of [2.5, 5.0, 7.5, 10, ...90, 100 cm], as well as time-series measurements of air temperature, precipitation and up- downward short- and long wave irradiation. Additionally, Matlab scripts for processing measured radar return data and for retrieving $\sigma_{pq}^0$ for other bands within the measured $1-10$ GHz frequency range are included in the dataset.

## 7 Conclusions

In this paper we describe a microwave scatterometer system that was installed on an Alpine Meadow over the Tibetan Plateau and its collected dataset consisting of measured radar returns from the ground surface. The observation period was August 2017 – August 2018 and measurements were taken with a one- to half hour temporal resolution. The scatterometer measured the radar return amplitude and -phase over a $1-10$ GHz band for all four linear polarization combinations. The system was built with commercially available components (vector network analyzer, four phase stable coaxial cables, and two broadband dual polarization gain horn antennas) and required little to no maintenance.

We described a procedure on how to retrieve the backscattering coefficients for all four linear polarization combinations from radar return measurements of a VNA-based scatterometer system with two fixed antennas operating over a broad frequency range ($1-10$ GHz). The typical effects resulting from the wide antenna radiation patterns were dealt with by using the narrow-beam approximation in combination with the mapping of function $G^2/R^4(x, y)$ over the ground surface, so that proper footprint positions and -areas, and incidence angle ranges could be derived. The incidence angle range was frequency-dependent and varied from $0-60°$ for L-band to $47-59°$ for X-band. Since spatial averaging was not possible frequency averaging was applied to reduce fading uncertainty. Bandwidths for averaging were selected with help of the Improved Integral Equation Model ($I^2EM$) for surface scattering.

Backscatter measurements of a rectangular metal plate and rotated metal dihedral reflectors were used as reference targets to calibrate the scatterometer for all polarization channels. Measurements of the incidence-angle dependence of $\sigma_{pq}^0$ for asphalt agreed with previous findings, thereby showing our $\sigma^0$ retrieval method to be accurate.

The uncertainty of our retrieved $\sigma^0$ can be divided in a known part estimated from fading- and systematic measurement uncertainty, and an unknown part due to low angular resolution of the used antennas. The known measurement uncertainty in $\sigma^0$ was estimated with an error model providing 66 % confidence intervals that are different over frequency bands, polarizations and

the overall level of the radar return. Extreme values for $\Delta\sigma^0$ were $\pm 1.5$ dB for S-band with hh polarization when the overall $\sigma^0$ level was highest (during summer) and $\pm 5.5$ dB for C-band with vh polarization when the overall $\sigma^0$ level was lowest (during winter).

Despite aforementioned uncertainty in $\sigma^0$ and the additional unknown uncertainty, we believe that the strength of our approach lies in the capability of measuring $\sigma^0$ dynamics over a broad frequency range, $1-10$ GHz, with high temporal resolution over a full-year period. Alternatively, instead of the retrieved $\sigma^0_{pq}$ the measured radar return in this dataset could be used in combination with a microwave scattering model to account for the angle-dependence of $\sigma^0_{pq}$.

On three days during summer the radar backscatter was measured for different angles in elevation and azimuth to quantify the angular dependence of $\sigma^0$ and to assess the ground surface homogeneity. Presented analysis on the angle-variation data of $\sigma^0$ showed wavelength- and polarization dependent scattering behaviour due to vegetation that is in accordance with theory and previous findings. Furthermore, these measurements indicated that the surface associated with the (fixed) footprint for the time-series measurements to be representative of its surroundings.

In the retrieved time-series of $\sigma^0_{pq}$ for L-band ($1.5-1.75$ GHz), S-band ($2.5-3.0$ GHz), C-band ($4.5-5.0$ GHz), and X-band ($9.0-10.0$ GHz) we observed characteristic changes or features that can be attributed to seasonal changes of the surface conditions. For example a fully frozen top soil, freeze-thaw changes in the top soil, emerging vegetation in spring, and drying of soil.

Further studies with the obtained dataset allows for in-depth analysis of diurnal changes of surface top-soil moisture dynamics during all periods within the year. Availability of backscattering data for multiple frequency bands allows for studying scattering effects at different depths within the soil and vegetation canopy during the spring and summer periods. Finally, combining scatterometer data with measured ELBARA-III radiometry data (Su et al., 2020) creates a complementary dataset that allows for in-depth study of the soil moisture and -temperature dynamics below, and at, the air-soil interface.

*Author contributions.* JH wrote this paper, installed and operated the scatterometer system, developed the data processing, $\sigma^0$ retrieval process, and performed the data analysis. RvdV advised in the experiment designs, $\sigma^0$ retrieval process and paper structure. XW, ZW and DZ, handled the China customs logistics, installed and operated the scatterometer system. On a regular basis they maintained the scatterometer system and the Maqu site. CvdT advised in the $\sigma^0$ retrieval process. JW and ZS conceptualized the experiment design. All co-authors commented and revised the paper.

*Competing interests.* All authors declare that there are no conflicts of interests

*Acknowledgements.* This work was supported in part by ESA ELBARA-II/III Loan Agreement EOP-SM/2895/TC-tc, the ESA MOST Dragon IV Program (Monitoring Water and Energy Cycles at Climate Scale in the Third Pole Environment), the Netherlands Organization for Scientific Research under Project ALW-GO/14-29, the National Natural Science Foundation of China (grant no. 41971033) and the Fundamental Research Funds for the Central Universities, CHD (grant no. 300102298307).

## List of Symbols

| | | |
|---|---|---|
| $A_{fp}$ | Surface area of the footprint. | $\mathrm{m}^2$ |
| $a$ | $a$ dimension of reference target frontal projection. | $\mathrm{m}$ |
| $\alpha$ | Angle between tower's vertical axis and the orthogonal projection of the line from antennas to a ground surface segment onto the plane formed by the tower's vertical axis and the antenna boresight direction line. See also Fig. 5. For antenna boresight line $\alpha = \alpha_0$. | ° |
| $BW$ | Bandwidth associated with $E_e$ or $\sigma^0$. | GHz |
| $b$ | $b$ dimension of reference target frontal projection. | $\mathrm{m}$ |
| $\beta$ | Angle between line from antennas to a ground surface segment and projection of that same line onto the plane formed by the tower's vertical axis and the antenna boresight direction line. See also Fig. 5. For antenna boresight line $\beta = \beta_0$. | ° |
| $c$ | Speed of light. | $\mathrm{m\,s^{-1}}$ |
| $D$ | Antenna aperture width. | $\mathrm{m}$ |
| $\Delta E_T$ | Temperature-induced radar return uncertainty. | $\mathrm{V\,m^{-1}}$ |
| $\Delta I_N$ | Uncertainty in $I_N$. | $\mathrm{W\,m^{-2}}$ |
| $\Delta K$ | Reference target measurement uncertainty. | $\mathrm{W\,m^{-2}}$ |
| $E_e$ | Magnitude of total electric field strength at the receive antenna, originating from the (surface) target. | $\mathrm{V\,m^{-1}}$ |
| $E_e^g$ | Same as $E_e$, superscript $g$ denotes time-domain filter, or gate, is applied. | $\mathrm{V\,m^{-1}}$ |
| $E_{cr}^g$ | Remnant of the transmit-to-receive antenna (direct )cross coupling. This quantity is measured with antennas aimed skywards, superscript $g$ indicates same time-domain filter, or gate, as with $E_e^g$ was used. | $\mathrm{V\,m^{-1}}$ |
| $E_b$ | Lowest measurable signal by scatterometer, or background value of $E_e$. | $\mathrm{V\,m^{-1}}$ |
| $E_0^{g0}$ | Magnitude of total electric field strength at the receive antenna, originating from the reference target. Superscript $g0$ denotes Time-domain filter, or gate, is applied. | $\mathrm{V\,m^{-1}}$ |
| $E_{b0}^{g0}$ | Background level of $E_0^{g0}$. Superscript $g0$ denotes same Time-domain filter, or gate, as with $E_0^{g0}$ is applied. | $\mathrm{V\,m^{-1}}$ |
| $\epsilon_0$ | Permittivity of vacuum (and by approximation that of air). | $\mathrm{F\,m^{-1}}$ |
| $\epsilon_{soil}$ | Effective relative permittivity of a soil, which is a mixture of dry soil, water, minerals, organic material etc. Includes both real and imaginary part component. | – |
| $G$ | Antenna gain as a function of angle with respect to antenna boresight direction. Maximum value is $G_0$. | – |
| $H_{ant}$ | Height of the antenna apertures above the ground. | $\mathrm{m}$ |

| | | |
|---|---|---|
| $I$ | Time-average intensity of total electric field strength at receive antenna, originating from the (surface) target. | $\mathrm{W\,m^{-2}}$ |
| $I_N$ | Measured intensity averaged over $N$ independent samples. | $\mathrm{W\,m^{-2}}$ |
| $\bar{I}$ | The average of a large amount of independent measurements of $I$ originating from a surface with backscattering coefficient $\sigma^0$. $\bar{I}$ is a surface property. | $\mathrm{W\,m^{-2}}$ |
| $K$ | Constant (over $BW$) linking $\sigma^0$ to $\bar{I}$ | $\mathrm{W\,m^{-2}}$ |
| $L$ | Maximum dimension of target in context of RCS measurement. | m |
| $M_v$ | Spatial average volumetric top soil moisture over Maqu site. | $\mathrm{m^3\,m^{-3}}$ |
| $m_v$ | Volumetric soil content. | $\mathrm{m^3\,m^{-3}}$ |
| $N$ | Number of independent scatterometer measurements, or samples, of a (surface) | $-$ |
| $P_p^{Rx}$ | Power received by radar or scatterometer. The subscript refers to the linear polarization direction (horizontal $h$ or vertical $v$) that is measured by the antenna. | W |
| $P_p^{Tx}$ | Power transmitted by radar or scatterometer. The subscript refers to the linear polarization direction (horizontal $h$ or vertical $v$) that is transmitted by the antenna. | W |
| $P_p^0$ | Power received by radar or scatterometer from calibration target. The subscript refers to the linear polarization direction (horizontal $h$ or vertical $v$) that is measured by the antenna. | W |
| $\phi$ | Azimuth, or horizontal rotation angle of antennas. | $^\circ$ |
| $R$ | Distance antennas to (area) target (segment). | m |
| $R_c$ | Distance antennas to calibration standard. | m |
| $R_{ff}$ | Distance from antennas beyond which the antenna far-field radiation region is defined. | m |
| $R_{fp}$ | Distance antennas to centre of footprint. | m |
| $R_{pw}$ | Distance from antennas beyond which the wavefront of transmitted radiation is considered planar. | m |
| $r_{sg}$ | Start of the time-domain filter, also known as gate. | m |
| $r_{eg}$ | End of the time-domain filter, also known as gate. | m |
| $\sigma_{pq}$ | Radar Cross Section (RCS). First subscript denotes polarization direction (horizontal $h$ or vertical $v$) of the scattered- and second denotes that of the incident radiation. | $\mathrm{m^2}$ |
| $\sigma_{min}$ | Minimum detectable radar cross section (RCS) by scatterometer given a certain distance to target $R$. | $\mathrm{m^2}$ |
| $\sigma_{pq}^0$ | Backscattering coefficient. The radar cross section (RCS) associated with a distributed target over a certain (physical) area. First subscript denotes polarization direction (horizontal $h$ or vertical $v$) of the scattered- and second denotes that of the incident radiation. | $(-)$ |
| $T_{soil}$ | Soil temperature. | $^\circ$C |
| $T_{encl}$ | Temperature inside VNA enclosure. | $^\circ$C |
| $\tau_g$ | Temporal width of the time-domain filter, also known as gate | s |
| $\tau_p$ | Temporal pulse width. | s |
| $W_{ant}$ | Separation distance between the two antenna apertures. | m |
| $w_g$ | Spatial width of the time-domain filter, also known as gate. | m |

## Appendix A: Images Maqu site at different seasons

In this section we present a set of photographs of the Maqu site taken at different seasons since the installation of the ELABRA-III in January 2016. These may give the reader a global indication of how the site phenology changes throughout the seasons. A more thorough and periodic set of photographs of the site was not taken unfortunately.

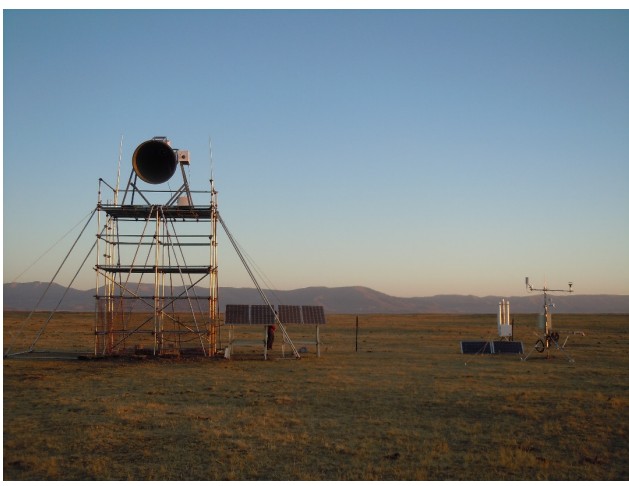

**Figure A1.** Winter, January 2016.

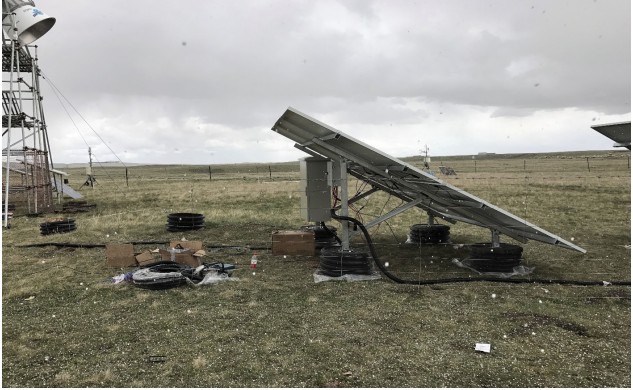

**Figure A2.** Spring, 16 May 2017.

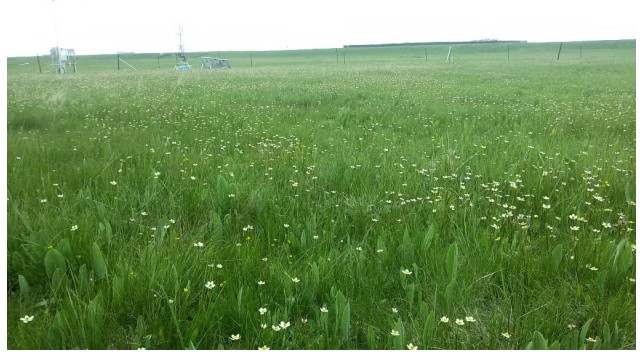

**Figure A3.** Spring, 26 June 2018.

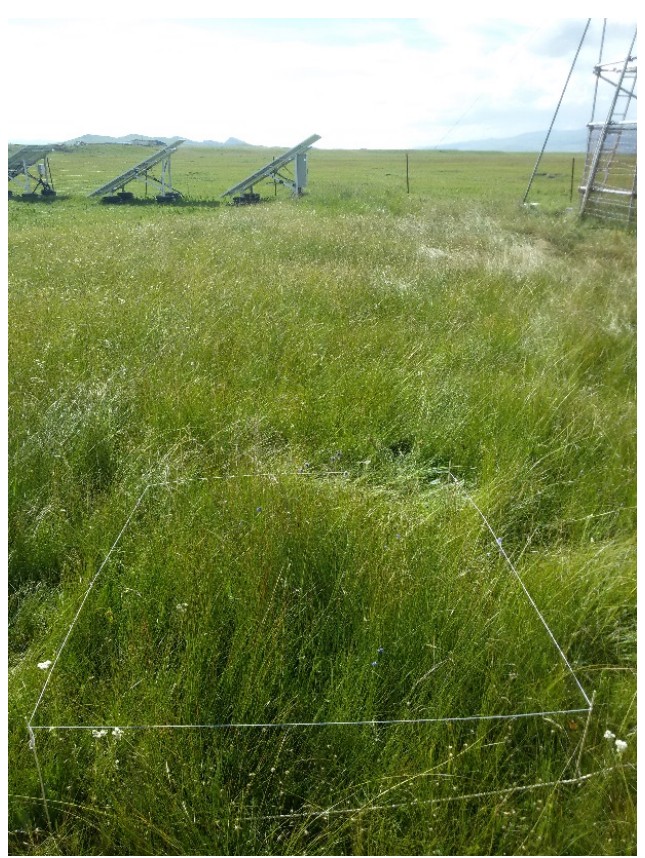

**Figure A4.** Summer, 17 August 2018.

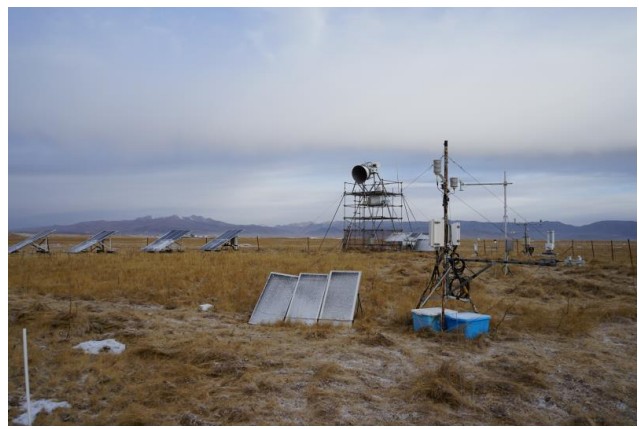

**Figure A5.** Winter, 6 January 2018.

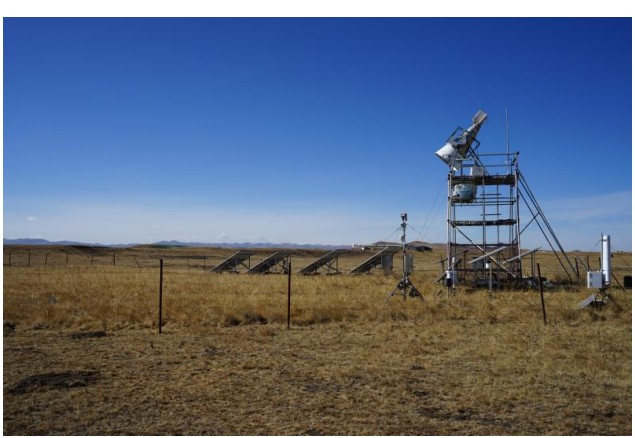

**Figure A6.** Winter, 6 January 2018.

## Appendix B:  Derivation spatial soil-moisture-variation estimate

At every depth, $m_v$ varies over the horizontal spatial extent at all scales (Famiglietti et al., 2008). Local $m_v$ variability is caused by variations in soil structure and texture, including organic matter. At the Maqu site, the 5TM sensor array forms only one spatial measurement point for soil moisture. We denote its measurements as $m_v^{5TM}$ ($\mathrm{m^3\,m^{-3}}$). In an attempt to quantify how $m_v^{5TM}$ at the top soil layer (depths 2.5 and 5 cm) relates to the soil moisture over the rest of the Maqu site, we sampled $m_v$ at 17 positions along the no-step zone (Fig. 3) on June $29^{th}$ 2018 with a hand held impedance probe, type ThetaProbe ML2x, whereby 3 measurements were taken per position. Figure B1 shows the measured $m_v$ in the top layer. Taking aside the outlying

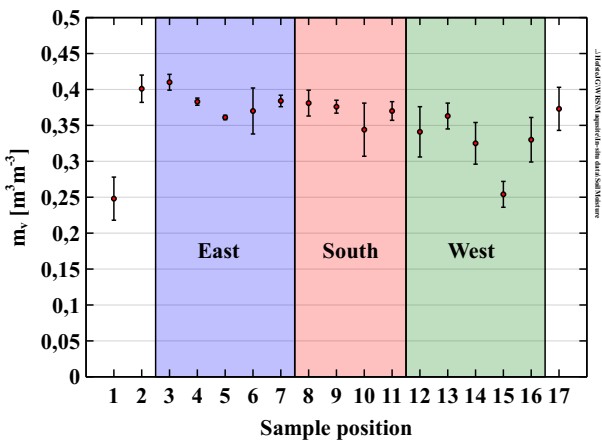

**Figure B1.** Top-soil $m_v$ measured with hand-held ThetaProbe at 17 sample positions along no-step zone periphery (indicated Fig. 3). Vertical bars denote minimum and maximum values of the 3 measurements per sample position. Red dots represent median values.

values at positions 1 and 15, we observe that the trend along the periphery is slightly larger than the variability amongst the three measurements taken at a specific position. The average standard deviation over the 15 positions is $0.03\ \mathrm{m^3\,m^{-3}}$ while the average standard deviation over the three measurements is $0.02\ \mathrm{m^3\,m^{-3}}$. Given this small difference we concluded there is no clear trend of top soil $m_v$ at the Maqu site. Therefore, we considered all $15 \times 3 = 45$ readings as independent measurements on spatial $m_v$ variation, that we used to create the quantity $S_{tot}$ ($\mathrm{m^3\,m^{-3}}$), called the total standard deviation of spatially measured $m_v$, which is an estimate for the spatial $m_v$ variability over the Maqu site. Subsequently, we use $S_{tot}$ to relate the measured $m_v^{5TM}$ to the spatial average top soil moisture content over the Maqu site $M_v$ ($\mathrm{m^3\,m^{-3}}$) according to

$$M_v = m_v^{5TM} \pm S_{tot} \tag{B1}$$

Using the assumption of temporal stability of spatial heterogeneity (Vachaud et al., 1985) we consider found $S_{tot}$ to hold throughout the year. $S_{tot}$ is calculated by

$$S_t = \sqrt{S_s^2 + S_{5TM}^2 + S_p^2} \tag{B2}$$

according to standard error propagation theory (see for example Hughes and Hase (2010)). The term $S_s$ ($\mathrm{m^3\,m^{-3}}$) represents the spatial $m_v$ variability as measured along the periphery. It is calculated as the standard deviation over $45-1$ samples and is 0.031 $\mathrm{m^3\,m^{-3}}$. The standard deviation $S_{5TM}$ a has value of 0.02 ($\mathrm{m^3\,m^{-3}}$) and is the root-mean-square measurement error of the 5TM sensors. It was derived in Zheng et al. (2017b) after calibrating 5TM sensor retrievals to top-soil gravimetric soil samples taken at the Maqu site. The term $S_p$ is the propagated error of the 0.05 $\mathrm{m^3\,m^{-3}}$ theta probe measurement accuracy (Table 2) when $S_s$ is calculated. $S_p = 0.05/\sqrt{45-1} = 0.0075\ \mathrm{m^3\,m^{-3}}$. Finally, $S_{tot}$ then is 0.04 $\mathrm{m^3\,m^{-3}}$.

## Appendix C: Details on scatterometer calibration

We measured the radar returns of reference targets with known radar cross section (RCS) $\sigma_{pq}$ in order to calibrate the scatterometer. For the co-polarization channels a rectangular metal plate was used as reference target. As a depolarizing reference target for the cross-polarization channels we used a metal dihedral reflector that was rotated 45° around the axis perpendicular to the vertex connecting the dihedral's two faces and contained in the symmetry plane also holding the same vertex. The physical optics model used for calculating the RCS of a metal plate and dihedral reflector is

$$\sigma_{pp} = 4\pi \frac{(ab)^2}{\lambda^2} \tag{C1}$$

where $a$ and $b$ are the standards' dimensions (m) in the frontal projection (Kerr, 1951). As is shown in for example (Nesti and Hohmann, 1990), Eq. C1 is also applicable for calculating the cross polarization RCS of the dihedral reflector when in its rotated position.

There are validity conditions for model C1 which concern the reference target's size and the distance at which it is measured $R_0$. Additionally, the multi-path field illumination of the reference targets (Skolnik, 2008) might be an issue: besides direct illumination from the transmit antenna, radiation reflected from the ground will also illuminate the target, see Fig. 5(b). As a result, the direct signal is interfered by these ground-to-target reflections. Table C1 lists the used $R_c$ values for the deployed reference standards. We first describe the validity conditions for model C1.

Conditions for Eq. (C1) are that the standard's largest dimension $L$ (m) is large compared to the wavelength, i.e. $L > \lambda$, and

**Table C1.** Deployed reference standards and their bandwidths of validity

| | Distance $R_0$ : | PW -criteria met for: | $L/\lambda \geq 3$ for: |
|---|---|---|---|
| Large rectangular plate, a = 85 cm, b = 65 cm | 36.3 m | $f \leq 7.5$ GHz | $f \geq 1.5$ GHz |
| Small dihedral reflector, a = 57 cm, b = 38 cm | 27.7 m | $f \leq 13$ GHz | $f \geq 2.4$ GHz |
| Large dihedral reflector, a = 120 cm, b = 65 cm | 27.7 m | $f \leq 3$ GHz | $f \geq 1.4$ GHz |

that the incident wavefront is close to planar. Kouyoumjian and Peters (1965) proposed the following equation for calculating the minimum distance $R_{pw}$ (m) beyond which the wavefront can be considered planar (allowing for a $\pi/8$ phase error):

$$R_{pw} = \frac{2L^2}{\lambda} \tag{C2}$$

Concerning the condition $L > \lambda$, previous measurements (Hofste et al., 2018) showed, empirically, that for $L/\lambda \geq 3$ model (C1) matches a standard's measured $\sigma_{pp}$ within 1 dB. Besides used $R_0$ values, Table C1 also lists the frequency ranges for which the plane wave criteria (using the stated values $R_0$) and the size criteria hold. Strictly speaking, the plane-wave criteria with the rectangular plate was not met for 7.5 - 10 GHz. Yet, the co-polarization $\sigma$ measurement of the small dihedral reflector, discussed in Sec. E2.2, yields results close to the Eq. C1 value, indicating correct values for 7.5 – 10 GHz.

Now we discuss the possible issue of multi-path illumination by ground-to-target reflections (GTR's). Should the signal strength of these GTR's be significant, the magnitude-over-frequency response of the reference targets will exhibit interference ripples, which complicate interpreting their radar return for the purpose of calibrating the scatterometer. By using gating the GTR's could in principle be removed from the direct target response, provided their difference in geometrical path length is large enough for placing a gating window solely over the direct path reflection in time domain. The GTR path shown in Fig. 5(b) was the pathway whose path length was closest to that of the direct route. Also, this GRT path will have the strongest coherent ground reflection since it is specular. Naturally, with smaller $R_0$ the difference $R_0 - (R_1 + R_2)$ increases, allowing one to better distinguish this GRT from the mean reflection.

However, no (clear) presence of any GRT could be found. Using a $BW = 0.5$ GHz bandwidth leads to a $\tau_p = 1/BW = 2$ ns resolution in the time-domain, which would allow us to see the shortest GTR-path reflection that -if present- should be at $[2R_c - (R_1 + R_2 + R_c)]/c = 5$ ns behind the direct-reflection peak. But even with S-band for hh-polarization (broad antenna pattern and for hh-polarization the coherent ground reflection is strongest) no GRT reflections could be found.

Because we could not find evidence of GRT interference we hypothesize that the GRT's were too small in magnitude for our case. The antenna patterns, certainly for the lower frequencies are broad enough to illuminate a large part of the ground surface, but because of the dense grass cover the coherent forward reflections were probably low. Additionally the bistatic RCS patterns of both the rectangular plate- and dihedral reflector are too narrow, even with L-band, for a sufficient amount of energy to be reflected (in a specular manner) back to the receive antenna. Typically the presence of interference due to multi-path illumination with setups like ours is tested by moving the reference target horizontally over a distance of half a wavelength and observing any changes in the signal. Unfortunately this procedure was not possible with our equipment.

## Appendix D: Gating

For simplicity, instead of using the (complex) electric field strength measured at the scatterometer's receive antenna $E_e$, we explain the gating process with the term $X$ (V), which can be considered proportional to $E_e$ by some scatterometer system constant. The measured frequency-domain signal $X[\omega_h]$ was transformed into the time-domain via the Inverse Digital Fourier Transform (IDFT), see for example (Tan and Jiang, 2013)

$$x[t_n] = \sum_{h=1}^{N} X[\omega_h] e^{i\omega_h t_n} \tag{D1}$$

$N$ is the total number of discrete frequency points within the bandwidth $BW$ (Hz) considered. Angular-frequency points $\omega_h$ (rad s$^{-1}$) and time points $t_n$ (s) are calculated with the minimum- and maximum frequency of $BW$, $f_{lo}$ and $f_{hi}$ respectively (Hz) via

$$\omega_h = 2\pi \left\{ [h-1] \left( \frac{f_{hi} - f_{lo}}{N-1} \right) + f_{lo} \right\} \qquad h = 1, 2, 3, ..., N \qquad \text{(D2)}$$

$$t_n = \frac{n-1}{f_{hi} - f_{lo}} \qquad n = 1, 2, 3, ..., N \qquad \text{(D3)}$$

Next the time-domain response $x[t_n]$ was multiplied by the time-domain filter, or gate, which was a block function of width $\tau_g$ whose sides fall off according to a rapidly decaying Gaussian function. The gate's start- and end times corresponded to the distances indicated in Fig. 5: $t_{sg} = 2r_{sg}/c$ and $t_{eg} = 2r_{eg}/c$ respectively. In this manner only the surface's scattering events of interest remained in the signal. Graphically, this is the intersection of depicted green ring of Fig. 5 and the scatterometer footprint $A_{fp}$. The gated signal $x[t_n]$ was then transformed back into the frequency domain via the Digital Fourier Transform

(DFT)

$$X[\omega_h] = \frac{1}{N} \sum_{n=1}^{N} x[t_n] e^{-i\omega_h t_n} \qquad \text{(D4)}$$

which then contains only the surface scattering information.

The frequency dependence of the radiation patterns, as shown in Fig. D1, complicates the process described above. The

785 time-domain equivalent of the transmitted scatterometer signal is a pulse of width $\tau_p = 1/BW$ s. Depending on the angle with respect to boresight, i.e. $\alpha$ & $\beta$, this signal pulse will contain different frequencies, and will therefore have a different temporal shape. At greater angles $\alpha$ & $\beta$, high-frequency components of the pulse are not present causing the pulse to be broader there. As a result, the footprint area $A_{fp}$, which is determined from the (known) antenna radiation- or gain patterns $G$ and the gate width $w_g = c\tau_g$ will become broader. We avoided this issue by narrowing our bandwidths such that the radiation patterns of

790 the frequencies within can be considered equal. As a consequence, this meant that for lower frequencies the selected $BW$ had to be more than those for the higher frequencies. Used bandwidths were 1.5 – 1.75 GHz for L-band, 2.5 – 3.0 GHz for S-band, for 4.5 – 5.0 GHz for C-band, for 9 – 10 GHz for X-band. Note that there were additional considerations for picking these $BW$ values, which are explained in Sec. 4.3.

When measuring the reference target backscatter responses $E_0$ (V m$^{-1}$) however, the full 0.75 – 10.25 GHz frequency range

can be used. Because the solid angle extending the standard is small we may reasonably assume that all frequencies are present in the time-domain equivalent pulse at the standard, i.e. $G(\alpha, \beta) \approx 1$ for all frequencies. The benefit of using this broad bandwidth (9.5 GHz) is a high temporal/spatial resolution in the time domain, which allows for precise placement of the gate over the reference target response.

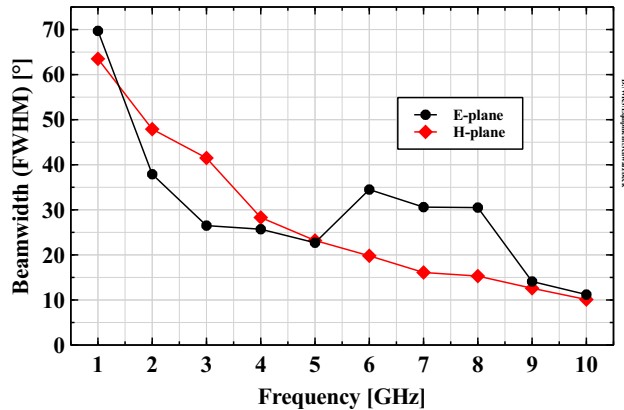

**Figure D1.** Beamwidths of dual polarization antennas. Shown is the full width half maximum (FWHM) of the measured radiation intensity patterns in the two principal planes (Schwarzbeck Mess-Elektronic, 2017).

## Appendix E: Details on sources of measurement uncertainty

### E1 Temperature-induced radar return uncertainty

The performance of the VNA's transmitters and receivers will vary due to variations of their operational temperatures, which in our case are directly linked to the temperature inside the VNA enclosure $T_{encl.}$. Many scatterometer systems employ a so-called internal calibration loop, see for example Ulaby and Long (2017), Baldi (2014), and Werner et al. (2010). This means that besides, or in between, scatterometer measurements the transmitter and receiver are connected, via a switch, through a reference transmission line of fixed length that has a pre-determined attenuation and phase. This way, any fluctuations in the transmitter and/or receiver output over time can be measured and consequentiality removed from the target response. Instead of such an internal calibration loop we employ a different method to account for temperature-induced fluctuations of the VNA's transmitter and receiver performance.

During a half-day timespan the antennas were aimed at a fixed target at 21 m distance: the bare metal mast (without the pyramidal absorbers in front) with on top a metal sphere. At half-hour intervals the radar return was measured together with $T_{encl.}$. The fixed target was assumed to remain constant during that time, so any changes in the radar return were attributed to the changing $T_{encl.}$, which varied from $25 - 35\ ^\circ$C during the experiment.

For bandwidths at L-band (1.50 – 1.75 GHz), S-band (2.5 – 3.0 GHz), C-band (4.5 – 5.0 GHz), and X-band (9.0 – 9.9 GHz) the radar returns $E_f$ (V m$^{-1}$) (subscript $f$ for 'fixed target') were filtered by a gate placed over the fixed target time-domain response, resulting in $E_f^{gf}$ (superscript $gf$ for 'gate over fixed target'). The change of $E_f^{gf}$ over time $t$, and thus over $T_{encl.}$, is denoted $\Delta E_f^{gf}(T_{encl.})$:

$$\Delta E_f^{gf}(T_{encl.}) = E_f^{gf}(t) - E_f^{gf}(t=0) \tag{E1}$$

In Fig. E1 the results of this experiment are shown. Plotted are the bandwidth-average difference of the S-parameter magnitudes over time (and temperature) with respect to the reference value $\Delta S_f^{gf}(T_{encl.})$, alongside with $T_{encl.}$. As explained in the main text, the quantities actually measured by the VNA were the S-parameters, which are proportional to the corresponding values $E_f^{gf}$ and $\Delta E_f^{gf}(T_{encl.})$.

There appeared to be no unique relationship between $\Delta S_f^{gf}$ and $T_{encl.}$. Within three hours from the experiment start $T_{encl.}$ increases to a maximum value after which it decreases again at an increasingly slowed rate. Also the curves $\Delta S_f^{gf}(T_{encl.})$, in general, change more rapidly over the first five hours and then become more stable. However, the direction of change in $T_{encl.}$: a rapid increase at the start, followed by a decrease after 19:15 at an increasingly slow rate is not seen in the $\Delta S_f^{gf}(T_{encl.})$ curves. So in order to quantify the temperature-induced VNA instability we used the maximum observed variation of $\Delta S_f^{gf}(T_{encl.})$

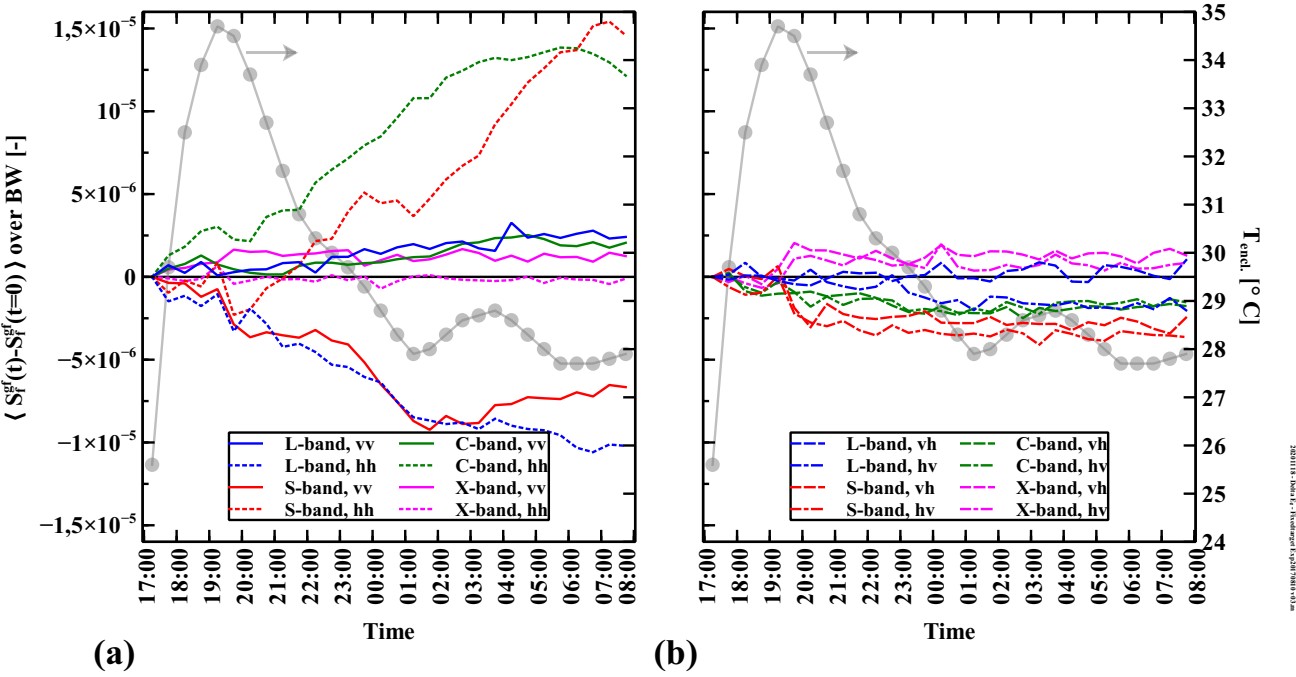

**Figure E1.** Measured radar return from a fixed target over a varying enclosure temperature $T_{encl.}$

over time amidst all frequencies within $BW$ to calculate the temperature-induced radar return uncertainty $\Delta S_T$. Or, in the context of scattered electric field strengths, its corresponding value $\Delta E_T$ (V m$^{-1}$)

$$\Delta E_T = \frac{max[\Delta E_f^{gf}(T_{encl.})] - min[\Delta E_f^{gf}(T_{encl.})]}{2} \tag{E2}$$

The quantity $\Delta E_T$ is to be treated as an absolute uncertainty of $E_e^g$ (Eq. 10) according to:

$$I_N = \frac{1}{2}c\epsilon_0 \frac{1}{N}\sum_{n=1}^{N}(E_e^g(f_n) - \langle E_{cr}^g \rangle - E_b \pm 2\Delta E_T)^2 \tag{E3}$$

with a factor two since both $E_e^g(f_i)$ and $E_b^g(f_n)$ are subject to this temperature-induced uncertainty. Table 4 lists the power levels at the VNA's receivers calculated from $\Delta S_T$ for the considered bandwidths and polarization channels.

## E2 Reference target measurement uncertainty

### E2.1 Reference target alignment

The absolute backscattering coefficient is determined with respect to the known RCS of a reference target. Errors in the used reference target's RCS itself, or errors made during the measurement of that target will contribute to the $\sigma^0$ uncertainty. The RCS of a rectangular metal plate calculated with Eq. (C1) was found to match experimental observations fairly well (Ross, 1966), and therefore errors in the RCS of our rectangular plate itself were not considered. For the dihedral reflector we do the same, keeping in mind that only the specular component was selected in time domain, thereby omitting interference from diffraction of the dihedral's edges. Should the gate have been wide enough to also cover these diffraction Eq. C1 will not be suitable anymore, see for example (Sorensen, 1991). We did consider errors in the measurement of the reference target, specifically we considered misalignment of the scatterometer's antennas towards the rectangular plate and vice versa.

The angular position of the reference targets with respect to the antenna boresight direction was estimated to be $-2.25° \leq \beta_0 \leq 1.25°$ in the horizontal direction and $-1.3° \leq \alpha_0 \leq 1.3°$ in the vertical direction. Given the large distance from the antennas to the rectangular plate, $R_0 = 36.3$ m, and the much smaller separation between the transmit- and receive antennas, $W_{ant} = 0.4$ m, single uncertainty values $\Delta\alpha_0$, $\Delta\beta_0$ were used for both antennas. Due to this possible antenna misalignment the reference target is not illuminated by the peak value of the gain pattern, i.e. $G = G(\alpha_0 \pm \Delta\alpha_0, \beta_0 \pm \Delta\beta_0)$ $(-)$, resulting in an uncertainty in the measured radar response of the reference target, and thus in $K$ (W m$^{-1}$). Equation 8 then is modified to

$$K = \frac{1}{2}c\epsilon_0(E_0^{g0} - E_{b0}^{g0} - E_b)^2 \frac{G(\alpha, \beta)^2}{G(\alpha_0 \pm \Delta\alpha_0, \beta_0 \pm \Delta\beta_0)^2} \left(\frac{R_0}{R_{fp}}\right)^4 \frac{A_{fp}}{\sigma^{bi}(\theta_i \pm \Delta\theta_i, \phi_i, \theta_s \pm \Delta\theta_s, \phi_s)} \tag{E4}$$

The angular position of the individual antennas with respect to the reference target's surface normal (or frontal projection surface normal in case of the dihedral reflectors) was estimated with the help of a laser mounted between the two antennas and detachable mirrors on the reference targets. Optimal alignment was found by rotating the targets until the reflected laser spot was on (or close to) the laser aperture again. In the horizontal plane, the angle between the rectangular plate's surface normal and the transmit antenna was $\theta_i = 0.16°$ (right side of the normal) and for the receive antenna $\theta_s = $ -0.48 °. In the vertical plane, the angle between the rectangular plate's surface normal and both antennas (as they are next to each other) was close to zero. We estimated the uncertainty of all aforementioned angles to be $\Delta\theta_i = \Delta\theta_s = 0.10°$ (both in the horizontal- and vertical plane.) For the small dihedral reflector these angles were $\theta_i = \theta_s = 0 \pm 0.2°$ in horizontal- and vertical plane while for the large dihedral reflector $\theta_i = 1.34 \pm 0.2°$ & $\theta_s = 0.52 \pm 0.2°$ in horizontal- and $\theta_i = \theta_s = 0.72 \pm 0.2°$ in vertical plane.

Starting with the physical optics model for the monostatic RCS of a metal rectangular plate, $\sigma(\theta, \phi)$ (Kerr, 1951) p. 457, a crude bistatic-RCS version $\sigma^{bi}(\theta_i, \phi_i, \theta_s, \phi_s)$ was created by simply imposing a linear phase delay along the plate's surface.

We shall assume that this model will also hold for the dihedral reflector. Calculation of $K$ can then be extended to include the (mis)alignment or offset of both individual antennas with respect to the reference targets and their uncertainties, which leads to Eq. E4.

How the uncertainties $\Delta\alpha_0$, $\Delta\beta_0$, $\Delta\theta_i$, and $\Delta\theta_s$ in Eq. E4 propagate into the uncertainty of $K$, called the reference target
measurement uncertainty $\Delta K$, may be found in textbooks such as Hughes and Hase (2010). Resulting $\Delta K$ values, per considered $BW$ and polarization, are presented as relative uncertainties in Table 4. With X-band the $\Delta K$ values are highest because the antenna radiation patterns are most narrow for higher frequencies.

### E2.2 Validation reference target alignment

In this section we shall demonstrate that estimated values for the rotational offsets and uncertainties $\theta_i$, $\theta_s$, $\Delta\theta_i$, $\Delta\theta_s$ of used reference targets are consistent with their respective measured radar returns. First we apply the radar equation (Eq. 3) to both the rectangular plate and the small dihedral reflector and substitute for $P^{Tx}$. We then have

$$\sigma_{dih}^{bi}(\theta_i^{dih},\phi_i,\theta_s^{dih},\phi_s) = \frac{P_{dih}^{Rx}}{P_{pla}^{Rx}}\frac{G(\alpha_0\pm\Delta\alpha_0,\beta_0\pm\Delta\beta_0)^2}{G(\alpha_0\pm\Delta\alpha_0,\beta_0\pm\Delta\beta_0)^2}\left(\frac{R_{dih}}{R_{pla}}\right)^4\sigma_{pla}^{bi}(\theta_i^{pla},\phi_i,\theta_s^{pla},\phi_s) \quad (E5)$$

where we dropped the polarization subscripts for readability. Since the values for $\alpha_0$ and $\beta_0$ are the same for both measurements
the term containing the antenna gain patterns $G$ is unity. We then end up with

$$\frac{\sigma_{dih}^{bi}(\theta_i^{dih},\phi_i,\theta_s^{dih},\phi_s)}{\sigma_{pla}^{bi}(\theta_i^{pla},\phi_i,\theta_s^{pla},\phi_s)} = \left(\frac{R_{dih}}{R_{pla}}\right)^4\frac{P_{dih}^{Rx}}{P_{pla}^{Rx}} \quad (E6)$$

Figure E2 shows the measured radar returns of the three calibration standards. For 5 GHz the difference between the small dihedral return $P_{dih}^{Rx}$ and the rectangular plate $P_{rect}^{Rx}$ for vv polarization is -3.3 dB. The term involving the distances $R$ is -4.7 dB resulting in the right-side of Eq. E6 to be -8.0 dB. If both reference targets were perfectly aligned towards the antennas the
885 RCS ratio on the left-side of Eq. E6 is -8.1 dB, which is 0.1 dB below the measured result. By finding suitable combinations of misalignment- or offset angles $\theta_i$, $\theta_s$ for both targets Eq. E6 can be satisfied. It can be shown that consistent angles can be found for all three reference targets which are within the ranges specified in section E2.1. In the above procedure we used the co-polarization returns of the dihedral reflectors, while it is in fact the cross-polarization that is of interest. The 45° rotation of the references for realizing the depolarization did not introduce significant other angular offsets. Note that the explained
method cannot validate the angular positions of the reference targets with respect to the antenna boresight direction and their uncertainties: $\alpha_0$ & $\Delta\alpha_0$ and $\beta_0$, $\Delta\beta_0$ as the term containing the antenna gain patterns was cancelled out.

We conclude this section with some remarks on the features in the measured reference target return powers shown in Fig. E2. With all returns there is a sharp trough between 8 – 9 GHz, which is caused by a combination of a local increment of the
895 antenna's return loss and an asymmetry in the antennas E-plane radiation pattern between 7 – 9 GHz. The asymmetry causes

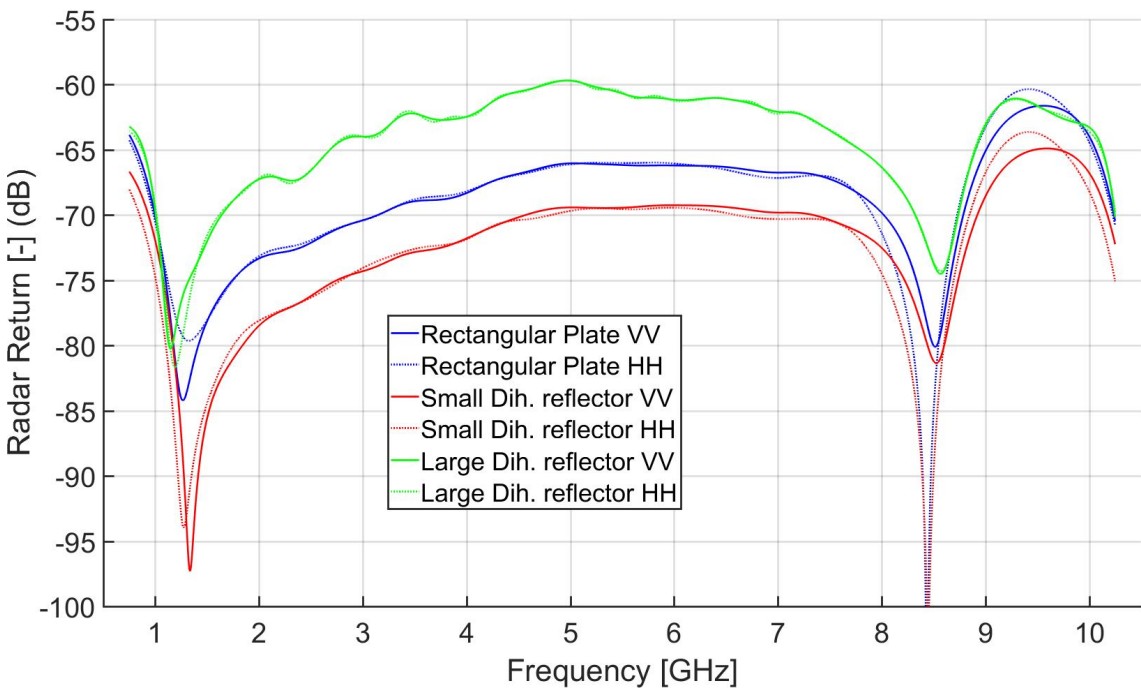

**Figure E2.** Measured radar returns of calibration standards for co polarization $E_0^{g0}$. Solid lines are VV- and dotted lines are HH polarization.

the pattern's peaks to point off-target by about $10°$ resulting in a lower radar return. The deep troughs close to 1.3 GHz are caused by a combination of high return loss at the low-frequency edge of the antenna's operational bandwidth and an artefact of the gating procedure, which in this case lets $E_0^{g0}(f)$ rise at the edge. This gating artefact is known to distort the band edges of a gated frequency response (Agilent, 2012). To account for this artefact the bandwidths used for the ground surface measure-
ments were broadened by 10% at both edges prior to gating. The added edges were discarded again after gating. The curves of the rectangular plate and small dihedral reflector have a similar shape for most of the frequency band. Their difference is merely a constant factor as predicted by the physical optics model for RCS (Eq. C1). The curve shape of the large dihedral reflector however is clearly different from the other two. This is partly because of its more severe angular offsets $\theta_i$ and $\theta_s$ but also because the planar-wave condition is not met for most of the frequency band, see Table C1.

### E3   Antenna coupling remnant

Because the transmit- and receive antennas are placed next to each other in order to measure the monostatic $\sigma^0$ part of the transmitted signal leaks, or couples, directly into the receive antenna, thereby interfering with the target return of interest. This antenna coupling is strongest for the lower frequencies (L-band) because these have the broadest antenna radiation patterns
(see Fig. D1). With respect to the polarization channels the antenna coupling is strongest for hh polarization because of how

the electric field lines of the principal TE$_{10}$ modes, in the particular case of hh polarization, couple strongest when the antenna apertures are next to each other. With the cross polarization channels the coupling is weakest because of how the principal field components are perpendicular between the transmit- and receive antenna.

Although the majority of the antenna coupling can be filtered out by gating, a remnant remains present in the filtered frequency domain response. This becomes apparent when the antennas are pointed skywards and the time-domain response is calculated per $BW$. Between the times/distances $r_{sg} = ct_{sg}/2$ and $r_{eg} = ct_{eg}/2$ where, during measurement of the ground target, the scattering events of interest are located the signal is not yet at its lowest level beyond 10 m. This effect is strongest for the L-band $BW$ with hh polarization while for X-band the time-domain response level between $r_{sg}$ and $r_{eg}$ is almost equal to its lowest level.

From the sky measurement the coupling remnant $E_{cr}^g(BW)$ was retrieved. When measuring the ground surface, the antenna coupling process of course interferes with the ground return. However, because we measure over a bandwidth and the ground return is a randomly fluctuating signal we argue that the $E_{cr}^g(BW)$ can simply be subtracted from the (gated) ground return $E_e^g(BW)$.

## E4   Propagataion of uncertainties

In this section we demonstrate how Eq. 12 is derived. We show, using error-propagation theory, how each of the (three) error-terms $\Delta E_T$, $\Delta K$, and fading, propagates into an error for $\sigma^0$ and how all errors may be combined into one statistical confidence interval for $\sigma^0$. We start with Eq. 6, which with Eq. 9 can be written as

$$\sigma^0 = \frac{\bar{I}}{K} = \frac{I_N}{K(1 \pm 1/\sqrt{N})} \tag{E7}$$

The term between brackets in the denominator we may simply rewrite as $F \pm \Delta F$, i.e. a variable with an error. The variables $I_N$ and $K$ also have their respective errors $\Delta I_N$ and $\Delta K$. When we write all variables and their errors explicitly we end up with

$$\sigma^0 = \frac{I_N}{KF} = \frac{I_N \pm \Delta I_N}{(K \pm \Delta K)(F \pm \Delta F)} \tag{E8}$$

We shall now describe all three error terms, starting with $\Delta I_N$. The calculation of $I_N$ from the measured backscattered electric field is given by Eq. E3 as

$$I_N = \frac{1}{2} c \epsilon_0 \frac{1}{N} \sum_{n=1}^{N} (E_e^g(f_n) - \langle E_{cr}^g \rangle - E_b \pm 2\Delta E_T)^2 \tag{E9}$$

with $\Delta E_T$ as measurement uncertainty. As explained in Sec. 4.3, every term in the above sum may be considered an independent variable. Because the number of samples $N$ within $BW$ is sufficiently large (about 15) we consider $\Delta E_T$ as a statistical error and therefore use the corresponding equation for error propagation (see for example Hughes and Hase (2010)) to calculate

the total statistical error $\Delta I_N$:

$$\Delta I_N = \frac{1}{2} c \epsilon_0 \frac{4 \Delta E_T}{N} \sqrt{\sum_{n=1}^{N} (E_e^g(f_n) - \langle E_{cr}^g \rangle - E_b)^2} \tag{E10}$$

$\Delta I_N$ can be considered as the one-standard-deviation value of $I_N$. Since the number of terms in the sum $N$ are large enough we can consider $\pm \Delta I_N$ as the edges of a 66 % confidence interval for $I_N$.

As explained in Sec. E2.1, $\Delta K$ can be calculated by using error propagation theory for the errors $\Delta \alpha_0$, $\Delta \beta_0$ and those associated with the bistatic RCS of the rectangular metal plate and dihedral reflectors $\Delta \theta_i$ and $\Delta \theta_s$. Note however that these are maximum possible errors so that the corresponding error propagation rules should be used. In order to have differentiable functions for the E-plane and H-plane antenna gain patterns, $E_{patt}(\alpha_0)$ and $H_{patt}(\beta_0)$ respectively, the measured radiation patterns can be fitted with Gaussian functions for angles close to antenna boresight. Writing $\Delta K$ explicitly is then straightforward.

Finally the error $\Delta F$, which of course is $1/\sqrt{N}$. As explained in Sec. 4.2 this error represents a 68% confidence interval for $\bar{I}$.

Returning to Eq. E8 we now combine all three errors into one statistical error. To do so we must first convert $\Delta K$ from being a maximum possible error into a statistical error like $\Delta I_N$ and $\Delta F$. This can be done by multiplying $\Delta K$ with 2/3, so the result may be interpreted as a one standard deviation value for $K$. This is equivalent to saying that $\pm 2/3 \Delta K$ is a 68 % confidence interval for $K$. We combine the three statistical errors conservatively into a 66 % confidence interval for $\sigma^0$:

$$\sigma^0 = \frac{I_N}{KF} = \frac{I_N \pm \Delta I_N}{(K \pm \frac{2}{3} \Delta K)(1 \pm 1/\sqrt{N})} = \frac{I_N}{KF} \pm \Delta \sigma^0 = \frac{I_N}{K} \pm \Delta \sigma^0 \tag{E11}$$

where $\Delta \sigma^0$ is calculated according to the error propagation equation for statistical errors:

$$\left( \Delta \sigma^0 \right)^2 = \left( \frac{\partial \sigma^0}{\partial I_N} \right)^2 (\Delta I_N)^2 + \left( \frac{\partial \sigma^0}{\partial K} \right)^2 (\Delta K)^2 + \left( \frac{\partial \sigma^0}{\partial F} \right)^2 (\Delta F)^2 . \tag{E12}$$

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
