# Peer review of "Year-long, broad-band, microwave backscatter observations of an Alpine Meadow over the Tibetan Plateau with a ground-based scatterometer"

_Earth System Science Data, 2020_

## Referee Comment (RC1) · Anonymous Referee #1 · 15 Jun 2020

The submitted manuscript deals with the acquisition, pre-processing as well as the introduction and explanation of a ground-based broad-band (1-10GHz), year-long (08/2017–08/2018), multi-polarimetric (HH,HV,VH,VV) scatterometer dataset from the Tibetian Plateau (Maqu, China) to study backscattering dynamics and their relation to environmental conditions at the alpine meadow site at hourly and half-hourly intervals. The hardware instrumentation, data acquisition and pre-processing is explained quite in detail, while the presentation of the dataset falls a bit short. But the dataset appears unique. Hence, the reviewer finds the biggest challenge in reasoning if the uniqueness

of the scientific dataset is presented in a complete and comprehensive way. Leading theme: What is the novelty of the dataset to our science community and is it shown adequately to advertise usage for research?

These are the reviewer's seeding questions for manuscript update on:

Completeness of dataset analysis/preparation and adequate presentation: -Why to confine the analysis/presentation to typical microwave bands? -What is about showing more broad-band/wide-band analyses? Can we find characteristics of the observed media (alpine meadow) in non-typical/non-standard microwave bands? -Why not adding at least L-band (if broad-band is not possible for some reasons)? -What about showing the variation in backscatter with viewing geometry (incidence: alpha & azimuth: Phi) also along time (similar to Figs. 9 & 10)? -Why not analyzing and showing all polarizations in analysis and presentation? -What is the benefit of polarimetry in this data set? -Why not showing at least cross-polarizations (e.g. in Fig. 13)? -Why soil temperature is mentioned as target variable? -Why not showing more of the time series of the data set? Can we correlate to seasons? Can we correlate to special (hydrological) events (e.g. drought)? -What are the exact ground conditions and their dynamics along the year? -Are there images how the site changes in phenology along the year (e.g. blooming or snowy or icy conditions)? -Is it possible to add further in situ time series curves (e.g. precipitation, vegetation conditions, solar irradiation . . .) to the figures to obtain a complete eco-hydrological picture of the alpine meadow? -Why not showing a temporal high-resolution freezing or thawing event with the hourly or half-hourly data? Just as a teaser for scientists to attract subsequent data usage.

Data preparation: -What about RFIs of the surrounding (area) and the hardware / measurement setup? -Is the radiometric calibration accuracy reported in the manuscript? -How were cross-polarized backscatter (HV, VH) measurements (pre-)processed? - Statement line 390: Is this correct? Does the wavelength double? -Where is the asphalt zone? Can we sketch it in Fig. 2?

Graphical presentation: -Why are figures kept small throughout the document (e.g. Figs. 7, 12 & 13)? -Which figure shows all the seasonal and diurnal changes of the different polarimetric backscatters along the year?

Citation: -Please add references to all equations not developed in the manuscript & to preceding research.

Outlook: -Why the scatterometer data is not combined with the radiometer data (ELBARA-III) for an active-passive combined dataset (in the future). This would be surely an even more unique dataset and fosters joint active-passive microwave research.

In a later stage of the manuscript (further) minor comments will be given.

––––––––––––––––––––––––

---

## Referee Comment (RC2) · Anonymous Referee #2 · 20 Jun 2020

The manuscript describes a dataset of backscatter observations of an Alpine Meadow with significant soil freezing-thawing processes, which includes measurements of both diurnal and seasonal cycles. I found this is a valuable study and dataset for Earth system sciences. It is suggested to accept it for publication after a minor revision.

Comments are listed as bellow.

Line 17: As the system is measuring data from 1 to 10 GHz at four linear polarization combinations, it is strongly suggested to provide time-series data at L-band with all four

polarizations, which will benefit the community a lot and future satellite missions operating at L-band, including the NASA-ISRO SAR (NISAR) mission [1] and the Terrestrial Water Resources Satellite (TWRS) [2]. Especially for the potential TWRS mission, it is aimed to measure the surface soil moisture and freeze/thaw state by the synergy use of active and passive observations at L-band. [1] Rosen, P. A., Kim, Y., Kumar, R., Misra, T., Bhan, R., & Sagi, V. R. (2017, May). Global persistent SAR sampling with the NASA-ISRO SAR (NISAR) mission. In 2017 IEEE Radar Conference (RadarConf) (pp. 0410-0414). IEEE. [2] Zhao, T., Shi, J., Lv, L., Xu, H., Chen, D., Cui, Q., ... & Zhao, K. (2020). Soil moisture experiment in the Luan River supporting new satellite mission opportunities. Remote Sensing of Environment, 240, 111680.

Line 109: Is the Maqu site a permafrost or a seasonal frozen ground area? Have you confirmed that all the soil depths would be thawed during the summer?

Line 219: should it be m2/m2?

Figure 8: Should the line in between be solid for the cyan lines (model simulations for HH)? Is that possible to include the S- and C-band also? How about the data at cross-polarizations?

Figure 9-10: It is suggested to include data and results from all four typical bands (L/S/C/X) in this Section, which would attract more interests.

Line 475: Are there any other observations to show it is snowfall, such as the camera, albedo etc.? It is better indicated of snow information in Figure 12. It is also suggested to indicate the date of soil freezing and thawing, as it seems to be the main target for this measurement as mentioned in the abstract.

Line 501-505: I am not very convinced by your argument. Even the longer wavelength will penetrate deeper into soil, the S-band should also be sensitive to the top-layer soil as the major contribution comes from the top soil, in which larger water phase transition (liquid to ice) occurs. We have conducted a similar multi-frequency observation by microwave radiometry over a seasonal frozen ground. It is very ingesting that brightness temperature and backscatter performed differently for the freezing-thawing process. Might be this is out of the scope of this data description paper; however, this is the value of presented measurements in this paper.

Line 515-516: Is that possible to process sigma0 for cross-polarization also? It would be more interested to share with the community with the processed sigma0 for all the four polarizations and typical bands (L/S/C/X).

To make it clear, I added references in my comments, not asking the authors specifically cite them but supporting my claims, these references could be cited in the manuscript only if authors deem it necessary.

---

## Author Response (AR2)

**ESSD-2020-44-RC1**

**"Year-long, broad-band, microwave backscatter observations of an Alpine Meadow over the Tibetan Plateau with a ground-based scatterometer"**

**Replies from Authors**

We, the authors would fist like to thank the first anonymous reviewer for taking the time to carefully read the manuscript and for providing comments on its contents and suggestions for its improvement. Please find below the replies of the authors to the comments (refering to version02) and, if applicable, performed actions (in blue) with corresponding lines in the new manuscript (verion04).

On behalf of all authors,

Jan Hofste.

Completeness of dataset analysis/preparation and adequate presentation:

**Comment** [1]: Why to confine the analysis/presentation to typical microwave bands?**

**Reply:** The authors are not sure what the reviewer means by 'typical' microwave bands? The bands we chose to show in the paper are: 2.5 - 3.0 GHz, from now on referred to as S'-band, 4.5 - 5.0 (C'-band), and 9.0 - 10.0 GHz (X'-band'). I shall explain why we chose these:

(1) Their respective wavelengths double each time approximately (see also reply [20]), thereby demonstrating, via the results, the wavelength-dependent scattering characteristics of the target. For example the effect of soil surface roughness and penetration depth of the soil and vegetation.

(2) We chose no more than three bands for the analysis to prevent the manuscript becoming too long.
(3) Instead of S', C', and 'X we could have chosen 'L, 'C, and 'X instead. As is explained in the document in section 5.1.2., the uncertainty of the absolute value of sigma0 will be largest for L-band where the antenna radiation patterns are widest. Although this is not a reason to discard this data, the interesting dynamics are unaffected, the authors chose S' instead to prevent this uncertainty from becoming too large. This way, the values presented in the paper can easily be compared to other studies.

We shall include the Matlab codes for processing the raw data in the dataset so that the readers can retrieve sigma0-timeseries according to their own preferences.

Action: Include Matlab codes for processing the raw data into sigma0-timeseries to online dataset. **Performed action:** Matlab codes included in dataset, (line 316).

**Comment [2]: What is about showing more broad-band/wide-band analyses?**

**Reply:** The authors are not sure what the reviewer means by broad-band? The bandwidths S', C' and X' used in the manuscript follow from a consideration where fading uncertainty is weighed against the frequency resolution of target's scattering response (see section 4.3) and the change of the antenna radiation patterns over frequency (see Appendix C, lines 681 - 690). Calculating sigma0 for a broad bandwidth of, for example, 2 - 4 GHz would imply averaging out any frequency-dependent effects and would lead to an additional uncertainty of the beam's footprint.

Performed action: Explanation added on chosen bandwidths in Section 4 (lines 250 – 255).

**Comment [3]:** Can we find characteristics of the observed media (alpine meadow) in non-typical/non-standard microwave bands?**

**Reply:** We assume here that the reviewer refers to the frequencies outside the bands S', C', and X' as used in the manuscript.

Microwave scattering and absorption are largely determined by the dimensions of the scattering elements with respect to the wavelength. Changing the wavelength only slightly will not alter the response very much. Suppose that in addition to S', C', and X' we also pick L' which spans 1.4 - 1.6 GHz. As explained under comment [1] the corresponding wavelengths approximately doubles between these bands and it is therefore likely to observe different scattering behaviour between them. The response of a 5th band, say at 6.4 - 6.9 GHz will probably not show as large a difference as between C' and X'. However, these 'intermediate bands' are useful as additional observations and can be used, for example, to decrease the fading uncertainty in sigma0.

**Comment [4]: Why not adding at least L-band (if broad-band is not possible for some reasons)?**

**Reply:** In reply [1] we explained why the response in L-band was not added into the manuscript. However, since the other reviewer also asked for L-band results we shall add these to the manuscript. **Action:** Add retrieved L-band sigma0 to manuscript in chapter 5.

**Performed action:** Retrieved sigma0 for L-band was added to all experiment results reported in Chapter 4 (line 249) and Appendix D (line 688). Also related descriptions regarding calibration were added in Section 3.2.2 (line 148) and in Appendices B3 (line 534).

**Comment [5]:** What about showing the variation in backscatter with viewing geometry (incidence: alpha & azimuth: Phi) also along time (similar to Figs. 9 & 10)?**

**Reply:** Although this would have been very useful from an experimental perspective -different spatial footprints would provide stronger evidence for the observed temporal changes of the measured surface- this was not possible with the used scatterometer setup. The setup was not equipped with an automated motorized rotational stage because this would make the setup vulnerable to instrument failures.

**Performed action:** Above explanation added to description experimental setup and -procedures section 3.2.2 (lines 167 - 171).

**Comment [6]: Why not analysing and showing all polarizations in analysis and presentation?**

**Reply:** To prevent the manuscript from becoming long we choose not to add the cross-polarization (X-pol) results. The measured X-pol data are included in the dataset, along with the measured response of a polarizing reference target for calibration. However, since also the other reviewer asked for adding the X-pol data into the manuscript we shall do so.

**Action:** Add retrieved sigma0 for the X-pol channels to chapter 5. Also in chapter 4 (derivation of sigma0) text will be added and/or modified so that the derivation of X-pol sigma0 is described. **Performed action:** Retrieved sigma0 for cross-polarization was added to all experiment results reported in Chapter 4 (line 249) and Appendix D (line 688). Also related descriptions regarding calibration were added in Section 3.2.2 (line 148) and in Appendices B3 (line 534).

**Comment** [7]: What is the benefit of polarimetry in this data set?**

**Reply:** The cross polarization channels could provide additional and complementary information on the target. Both for the vegetation and soil.

**Comment [8]:** Why not showing at least cross-polarizations (e.g. in Fig. 13)? **Reply:** See reply question [6].

**Comment [9]: Why soil temperature is mentioned as target variable?**

**Reply:** The strength of the microwave (back)scattering from the soil is dependent on the roughness and dielectric contrast of the air-to-soil interface. The soil temperature is mentioned because it is one of the variables that effects the effective dielectric permittivity of the bare-soil-and-water mixture particularly in the transition from thawing to freezing.

Comment [10]: Why not showing more of the time series of the data set?

**Reply:** Considering the length of the manuscript we chose not to present more measured data. Also the guides for ESSD data papers state that "Articles in the data section may pertain to the planning, instrumentation, and execution of experiments or collection of data. Any interpretation of data is outside the scope of regular articles." However, given the general remark by the reviewer questioning whether the dataset coverage is adequately demonstrated we will add more  $\sigma^0$  timeseries results, similar to the figure 13, to demonstrate the contents of the dataset. We shall show the response during the different seasons.

Action: Add more figures similar to figure 13 showing half-hourly sigma0 timeseries over consecutive days during all seasons.

**Performed Action:** In total four figures showing 13-day periods of retrieved half-hourly simga0 values are in the manuscript. Each for one season. See Chapter 4, figure 6 (line 307) and figures D1, D2, and D3 of appendix D (line 688).

**Comment** [11]: Can we correlate to seasons?**

**Reply:** In the first half of section 5.2.3, together with figure 12, a few general remarks are made considering the difference between the summer and winter period. So yes, we can correlate the scatterometer observations to the changing seasons. But, as explained with question [10] detailed analysis of the measured data is beyond the scope of an ESSD data paper.

**Comment** [12]: Can we correlate to special (hydrological) events (e.g. drought)?**

**Reply:** There were no long droughts during the timeseries considered (August 2017 – August 2018). However, there are periods during summer, for example in July, when there is little to no precipitation for 10 days or more during which the backscatter decreases in a different manner for S-, C- and X-band. This is due to the effects of vegetation. As mentioned under comment [10], we will a half-hourly sigma0 timeseries including such an event.

Action: Add a figure similar to figure 13 showing half-hourly  $\sigma^0$  timeseries during a number of consecutive days in summer 2018 when there was no rain. Performed action: See performed action on comment [10].

**Comment [13]: What are the exact ground conditions and their dynamics along the year?**

**Reply**: Time series data of volumetric soil moisture content and soil temperatures (at depths 2.5, 5, 7.5, 10 cm, ...) are included in the dataset. As are timeseries of precipitation. Besides these, also the air temperature, and incident- and reflected solar short- and longwave irradiation were measured at the Maqu site. To give the reader a better overview, graphs showing aforementioned hydrometeorological quantities will be added to chapter 2.

Action: Add figures in section 2.2 showing time-series measurements of volumetric soil moisture content, soil temperature, air temperature, precipitation, and incident- and reflected solar short- and longwave irradiation (also albedo).

**Performed action:** Section concerning supporting measurements was updated, this is now section 3.1 (line 109) and Appendix A2 (line 448). Figure A2 (line 467) shows volumetric soil moisture, soil temperature, air temperature, precipitation and daily total energy sum. The data of aforementioned quantities is also added to the dataset.

**Comment [14]:** Are there images how the site changes in phenology along the year (e.g. blooming or snowy or icy conditions)?

**Reply:** Apart from the measured incident- and reflected solar short- and longwave irradiation (which can be used to deduce snowfall) and the vegetation samples taken during two days in the 2018 summer, no periodical images of the site were taken. There are however several photographs taken of the site at different seasons of the year. These photographs we will add to the manuscript. **Action:** Add existing photographs of the Maqu site that give the reader an indication of the changing

Action: Add existing photographs of the Maqu site that give the reader an indication of the changing phenology over the year.

**Performed action:** In Appendix A (line 445) photographs of the Maqu site during different times of year are added. The photographs were not taken in the same year unfortunately, but they should provide the reader with a basic indication on the site phenology.

**Comment [15]:** Is it possible to add further in situ time series curves (e.g. precipitation, vegetation conditions, solar irradiation : : :) to the figures to obtain a complete eco-hydrological picture of the alpine meadow?

Reply: See reply of comments [13] and [14]

**Comment [16]:** Why not showing a temporal high-resolution freezing or thawing event with the hourly or half-hourly data? Just as a teaser for scientists to attract subsequent data usage. **Reply:** Figure 13 shows exactly this. Although detailed information is lost due to the small size of the figure.

Action: Increase size of figure 13.

Performed action: Size of this particular figure (figure 6, line 305), and similar figures is increased.

**Data preparation:**

**Comment [17]:** What about RFIs of the surrounding (area) and the hardware / measurement setup? **Reply:** Any possible RFI signals, capable of reaching the receiver's final stage, were considered as part of the noise floor. This noise floor was determined by measuring the response with the antennas pointed skywards. In the retrieval of sigma0 this noise floor is subtracted from the target's signal. See section 5.1.1 Table 4 and Appendix D4.

**Additional info:** In the new document RFI considered part of background level  $E_b$ , see section 6.2 (line 331).

**Comment [18]:** Is the radiometric calibration accuracy reported in the manuscript? **Reply:** Yes. See section 5.1.1, Table 4 ( $\Delta$ K) and Appendix D2. **Additional info:** In the new document refer to section 6.2 (line 627) and Appendix E3 (line 756).

**Comment [19]:** How were cross-polarized backscatter (HV, VH) measurements (pre-)processed? **Reply:** See reply on question [6].

**Comment [20]:** Statement line 390: Is this correct? Does the wavelength double? **Reply:** between the used bandwidths 5 - 4.5 GHz and 3 - 2.5 GHz it is true that the wavelengths are not exactly doubled, only approximately: (6.0 - 6.7 cm) and (10 - 12 cm) respectively. **Action:** State that the wavelengths are approximately doubled. **Performed action:** Statement removed (line 250).

**Comment [21]:** Where is the asphalt zone? Can we sketch it in Fig. 2?

**Reply:** The backscatter measurements on asphalt were performed in the Netherlands earlier in 2017. The experimental setup and equipment used was the same as used at the Maqu site.

Graphical presentation:

**Comment [22]:** Why are figures kept small throughout the document (e.g. Figs. 7, 12 & 13)? **Reply:** The authors agree that the Figures are too small. **Action:** Increase figure size to maximum allowed size. **Performed action:** All figures in manuscript are increased.

**Comment [23]:** Which figure shows all the seasonal and diurnal changes of the different polarimetric backscatters along the year?

**Reply:** Figure 12 shows the seasonal changes of the co-polarization backscattering channels for the whole August 2017 – August 2018 period. As mentioned with comment [10], figures will be added showing the diurnal changes of sigma0 (for co- and cross polarization) during 13 days for all seasons (like figure 13).

**Citations:**

**Comment [24]:** Please add references to all equations not developed in the manuscript & to preceding research.

**Reply:** We shall add references to preceding research with those equations not developed in the manuscript.

Action: Add references to the equations not developed in the manuscript to preceding research. **Performed action:** Reference added with eq. (1) (line 181) and eq. A2 (Markup document, line 485). All other equations either already had references or were developed in manuscript.

**Outlook:**

**Comment [25]:** Why the scatterometer data is not combined with the radiometer data (ELBARA-III) for an active-passive combined dataset (in the future)? This would be surely an even more unique dataset and fosters joint active-passive microwave research.

**Reply:** This manuscript specifically concerns the scatterometer system and the processing of its gathered dataset. The ELBARA-III dataset is made available elsewhere by Su et al. [1]. The data paper accompanying this dataset is currently under review in Scientific Data. We will refer to this in the revised manuscript.

[1] Su, Z., Wen, J., Zeng, Y., Zhao, H., Lv, S., van der Velde, R., Zheng, D., Wang, X., Wang, Z., Schwank, M., Kerr, Y., Yueh, S., Colliander, A., Qian, H., Drusch, M., and Mecklenburg, S.: Multiyear in-situ L-band microwave radiometry of land surface processes on the Tibetan 1185 Plateau, Scientific Data, 7, 317, 2020.

**ESSD-2020-44-RC2**

**"Year-long, broad-band, microwave backscatter observations of an Alpine Meadow over the Tibetan Plateau with a ground-based scatterometer"**

**Replies from Authors**

We, the authors, would first like to also thank the second anonymous reviewer for taking the time to carefully read the manuscript and for providing comments on its contents and suggestions for its improvement. Please find below the replies of the authors to the comments (refering to version02) and, if applicable, performed actions (in blue) with corresponding lines in the new manuscript (verion04).

On behalf of all authors,

Jan Hofste.

**Comment [1]:** Line 17: As the system is measuring data from 1 to 10 GHz at four linear polarization combinations, it is strongly suggested to provide time-series data at L-band with all four polarizations, which will benefit the community a lot and future satellite missions operating at L-band, including the NASA-ISRO SAR (NISAR) mission [1] and the Terrestrial Water Resources Satellite (TWRS) [2]. Especially for the potential TWRS mission, it is aimed to measure the surface soil moisture and freeze/thaw state by the synergy use of active and passive observations at L-band.

[1] Rosen, P. A., Kim, Y., Kumar, R., Misra, T., Bhan, R., & Sagi, V. R. (2017, May). Global persistent SAR sampling with the NASA-ISRO SAR (NISAR) mission. In 2017 IEEE Radar Conference (RadarConf) (pp. 0410-0414). IEEE.

[2] Zhao, T., Shi, J., Lv, L., Xu, H., Chen, D., Cui, Q., ... & Zhao, K. (2020). Soil moisture experiment in the Luan River supporting new satellite mission opportunities. Remote Sensing of Environment, 240, 111680.

**Reply:** The bands we choose to show in the paper are: 2.5 - 3.0 GHz, from now on referred to as S'band, 4.5 - 5.0 (C'-band), and 9.0 - 10.0 GHz (X'-band'). We choose to not show more than three bands to prevent the manuscript becoming too long. Instead of S', C', and 'X we could have chosen 'L, 'C, and 'X instead. As is explained in the document in section 5.1.2. the uncertainty of the absolute value of sigma0 will be largest for L-band where the antenna radiation patterns are widest. Although this is not a reason for discarding this data, the interesting dynamics are unaffected, the authors choose S' to prevent this uncertainty from becoming too large. This way, the values presented in the paper can easily be compared to other studies. However, since also the other reviewer asked for the Lband retrievals to be added in the manuscript we shall do so.

The retrieved sigma0 for cross-polarization (X-pol) was also not included in the initially submitted manuscript to prevent the manuscript becoming too long. In the revised manuscript, however, we shall add the X-pol timeseries of sigma0 and the X-pol results for the asphalt measurements.

Finally we shall add the Matlab code for calculating sigma0 from the raw data to the online dataset so that the reader can retrieve sigma0-timeseries according to their own preferences.

Action: Add retrieved sigma0 timeseries for L-band to the manuscript in chapter 5. Also add retrieved sigma0 for the X-pol channels to sections 5.1 (uncertainty in sigma0), 5.2.1 (backscattering of asphalt), and 5.2.3 (Time-series of sigma0 Maqu). In chapter 4 (derivation of sigma0) text will be added and/or modified so that the derivation of X-pol sigma0 is also described. Include Matlab code of sigma0 retrieval from raw data to online dataset.

**Performed actions:** retrieved sigma0 for L-band and cross-polarization channels was added to all experiment results reported in Chapter 4 (line 250) and Appendix D (line 688). Also related descriptions regarding calibration were added in Section 3.2.2 (line 148) and in Appendices B3 (line 535).

**Comment [2]:** Line 109: Is the Maqu site a permafrost or a seasonal frozen ground area? Have you confirmed that all the soil depths would be thawed during the summer?**

**Reply:** The Maqu site is a seasonal frozen ground area. In the summer the soil at all depths is thawed eventually. Moreover, measurements of the soil temperature over investigated period showed no temperature below 0 °C at 70 cm depth and beyond. (line 109 - 110). This phenomena is also visible in the included soil moisture- and temperature dataset. We can add a more general overview of hydrometeorological conditions throughout the year in section 2.2 by adding a figure showing the soil moisture and -temperature, air temperature, precipitation and incident- and reflected short- and long wave irradiation (and albedo) over time.

Action: Add figures in section 2.2 showing time-series measurements of volumetric soil moisture content, soil temperature, air temperature, precipitation, and incident- and reflected solar short- and longwave irradiation (and albedo).

**Performed action:** Section concerning supporting measurements was updated, this is section 3.1 now (line 109) andAppendix A2 (line 449). Figure A2 (line 467) shows volumetric soil moisture, soil temperature, air temperature, precipitation and daily total energy sum. The data of aforementioned quantities is also added to the dataset.

Comment [3]: Line 219: should it be m2/m2? Reply: Indeed, the wrong units were placed. Action: Change to m2/m2. Performed action: Changed (Line 180).

**Comment [4]:** Figure 8: Should the line in between be solid for the cyan lines (model simulations for HH)? Is that possible to include the S- and C-band also? How about the data at cross-polarizations?

**Reply:** The in-between cyan line, indicating the mean value of the empirical model, should indeed be solid. This will be adjusted.

For the asphalt measurements we showed only the X-band data because only for that band we found multiple other studies [1] to compare our results to. The only other study on asphalt backscattering known to us (for bands within our measured 1—10 GHz range) is that of Baldi, 2014 [2]. However, we realize now that the absence of (multiple) other studies to compare our results to is no valid reason not to simply show ours. We shall show our measured results for L-, S,- C-, and X-band for all four linear polarization combinations.

Action: Adjust the cyan line in figure 8. Add the measurement results for L-, S-, and C-band with all four linear polarization combinations.

[1] Ulaby, F. and Dobson, M.: Handbook of Radar Scattering Statistics for Terrain, Artech House Inc., Norwood MA, USA, 1989.

[2] Baldi, C.: The design, validation and analysis of surface based S-band and D-band polarimetric scatterometers, Thesis, 2014.

**Performed action:** The section on angle-dependent backscattering for asphalt (Appendix F, line 881) was updated to include all four bands and polarizations. A brief analysis and (were possible) comparisons with previous studies were added.

**Comment [5]:** Figure 9-10: It is suggested to include data and results from all four typical bands (L/S/C/X) in this Section, which would attract more interests.

**Reply:** As mentioned in the answer to comment [1] analysis for L-band will be added here as well. **Action:** Add analysis for L-band section 5.2.2.

**Performed action:** Section on the angular variation of sigma0 in Maqu was updated to include Lband and cross polarization data, section 6.3 (Line 364) and Appendix G (Line 921)

**Comment [6]:** Line 475: Are there any other observations to show it is snowfall, such as the camera, albedo etc.? It is better indicated of snow information in Figure 12. It is also suggested to indicate the date of soil freezing and thawing, as it seems to be the main target for this measurement as mentioned in the abstract.

**Reply:** Daily photographs of the site were not taken unfortunately. As indicated in reply [2] an overview of the hydrometeorological parameters and incident- and reflected short- and long wave irradiation (and albedo) will be added in section 2.2.

Action: Add figures in section 2.2 showing time-series measurements of volumetric soil moisture content, soil temperature, air temperature, precipitation, and incident- and reflected solar short- and longwave irradiation (and albedo). Also add figure showing timeseries of sigma0 during winter at maximum temporal resolution (like Figure 13). Indicate snow events in this figure.

**Performed action:** See performed actions of comment [2]. Additionally, in Appendix A (line 441) photographs of the Maqu site during different times of year are added. The photographs were not taken in the same year unfortunately, but they should provide the reader with a basic indication on the site phenology. In figures 5 (line 289), 6 (line 307), and figure A2 (line 467) snowfall events are indicated. Soil freezing/thawing is indicated in figure 5, 6 and figure A2.

**Comment [7]:** Line 501-505: I am not very convinced by your argument. Even the longer wavelength will penetrate deeper into soil, the S-band should also be sensitive to the top-layer soil as the major contribution comes from the top soil, in which larger water phase transition (liquid to ice) occurs. We have conducted a similar multi-frequency observation by microwave radiometry over a seasonal frozen ground. It is very interesting that brightness temperature and backscatter performed differently for the freezing-thawing process. Might be this is out of the scope of this data description paper; however, this is the value of presented measurements in this paper.

**Reply:** It is possible that the argument for explaining the differences in diurnal change of sigma0 for the different bands is indeed invalid. More analysis is necessary to find a satisfying explanation for the observed phenomena. As the reviewer suggests, this analysis is outside the scope of this data paper.

Action: Remove current explanation from manuscript and add sentence "In general the magnitude of the sigma0- change ..." to preceding paragraph.

Performed action: Explanation is removed (Line 304).

**Comment [8]:** Line 515-516: Is that possible to process sigma0 for cross-polarization also? It would be more interested to share with the community with the processed sigma0 for all the four polarizations and typical bands (L/S/C/X).

**Reply:** This comment is related to [1]. Yes, we shall process the cross-polarization data for already considered bands (S',C',X') and also for L-band.

Action: Add processed L-band data (co- and cross polarization) and cross polarization data for S', C', and X' to dataset.

**Performed action:** Processed sigma0 for L-, S-, C-, and X-band, with all 4 polarization channels is added to dataset (line 311). Cross-polarization results also added in document in Chapter 4 (line 249), and section 6.3 (line 365).

---

## Author Response (AR3)

Jan Hofste 2021 04 19

By request of the topical editor a minor change was applied to the manuscript.

Change: DOI links in manuscript to the dataset and Matlab code corrected.

Old version: essd-2020-44-manuscript-version4.pdf

New version: essd-2020-44-manuscript-version5.pdf